# Rational design of a triple-type human papillomavirus vaccine by compromising viral-type specificity

Zhihai Li[1], Shuo Song[1], Maozhou He[2], Daning Wang[2], Jingjie Shi[1], Xinlin Liu[1], Yunbing Li[1], Xin Chi[2], Shuangping Wei[1], Yurou Yang[1], Zhiping Wang[1], Jinjin Li[2], Huilian Qian[2], Hai Yu[1], Qingbing Zheng[1], Xiaodong Yan[2,3], Qinjian Zhao[2], Jun Zhang[2], Ying Gu[1,2], Shaowei Li [1,2] & Ningshao Xia [1,2]

Sequence variability in surface-antigenic sites of pathogenic proteins is an important obstacle in vaccine development. Over 200 distinct genomic sequences have been identified for human papillomavirus (HPV), of which more than 18 are associated with cervical cancer. Here, based on the high structural similarity of L1 surface loops within a group of phylo-genetically close HPV types, we design a triple-type chimera of HPV33/58/52 using loop swapping. The chimeric VLPs elicit neutralization titers comparable with a mix of the three wild-type VLPs both in mice and non-human primates. This engineered region of the chimeric protein recapitulates the conformational contours of the antigenic surfaces of the parental-type proteins, offering a basis for this high immunity. Our stratagem is equally successful in developing other triplet-type chimeras (HPV16/35/31, HPV56/66/53, HPV39/68/70, HPV18/45/59), paving the way for the development of an improved HPV prophylactic vaccine against all carcinogenic HPV strains. This technique may also be extrapolated to other microbes.

[1] State Key Laboratory of Molecular Vaccinology and Molecular Diagnostics, School of Life Sciences, Xiamen University, Xiamen, China 361102. [2] National Institute of Diagnostics and Vaccine Development in Infectious Disease, School of Public Health, Xiamen University, Xiamen, China 361102. [3] Department of Chemistry and Biochemistry and Division of Biological Sciences, University of California-San Diego, San Diego, CA 92093-0378, USA. These authors contributed equally: Zhihai Li, Shuo Song, Maozhou He. Correspondence and requests for materials should be addressed to Y.G. (email: guying@xmu.edu.cn) or to S.L. (email: shaowei@xmu.edu.cn) or to N.X. (email: nsxia@xmu.edu.cn)

Vaccines are highly efficient weapons against infectious disease. However, multiple antigenic types or subtypes derived from the evolution of pathogenic microbes through sequence variation presents a serious obstacle in vaccine development. One way to tackle this variation is to include more antigenic variants into a single vaccine, as exemplified with the *Streptococcus pneumoniae* vaccine[1] and *Human papillomavirus* (HPV) prophylactic vaccine[2]. Yet, because pathogens, such as the influenza viruses and human immunodeficiency virus (HIV), have very high levels of antigenic variability, this approach is fraught with difficulties, as an increase in type coverage will dramatically enhance protein amount and adjuvant level per dose, as well as increase the manufacturing complexity and associated production costs. Studies that focus on designing immunogens capable of inducing a broader protection against multiple subtypes or variants require technical methods, such as computationally optimized broadly reactive antigen (COBRA)[3], which uses the consensus sequence from multiple variants to increase the immunogenicity of the conserved epitopes that are shared between subtypes and targeted by broadly neutralizing antibodies among subtypes[4–6]. As yet, however, few studies have been successful, and there is thus an urgent need to identify or design antigens that can elicit antibodies with high and broad anti-virus potency.

Oncogenic HPV infection is associated with several malignancies, including cervical and anogenital cancer[7]. To date, more than 200 distinct HPV genotypes have been identified, of which at least 18 belong to the "high-risk" group and are chiefly responsible for the development of cancer[8–10]. HPVs are non-enveloped, double-stranded DNA viruses comprising multiple copies of the major (L1) and minor (L2) capsid proteins. The native $T = 7$ HPV virion can be mimicked by an empty icosahedral shell, called a virus-like particle (VLP) which consists of 72 L1-only pentamers[11,12]. High-level neutralizing antibodies elicited by L1 VLPs can block HPV infection; however, protection against HPV infection is type-restricted, and there tends to be little cross-reactivity among HPV types[13–15]. Pre-clinical and clinical data show that vaccinated patients exhibit low titers of neutralizing antibodies against genetically related, non-vaccine HPV types in highly reactive immune sera, and these antibodies may drop below protective levels sooner than type-specific ones[16–18]. For these reasons, the current market-available prophylactic HPV vaccines: Cervarix[19], Gardasil[20] and Gardasil 9 are multivalent vaccines of the various HPV L1 proteins. For example, Gardasil 9 provides type-specific protection against infection from nine HPV types (HPV6, -11, 16, -18, -31, -33, -45, -52 and -58), the last seven of which are responsible for about 90% of cervical cancers.

The evolutionary history of HPV indicates a slow rate of HPV mutation[21]; however, it is possible that genetic mutation and recombination between different viruses may result in vaccine evasion; for example, a recombination event between HPV16 and HPV31 may result in a virus carrying HPV16 oncogenes but coding for HPV31 structural proteins[21]. Indeed, previous work shows that substitution of just a few residues on the FG loops of L1 proteins between HPV16 and HPV31 can create a new serotype, which would avoid the immunity conferred by both types[22]. Furthermore, it remains unclear whether widespread immunization with vaccines like Gardasil 9 would lead to an increase in infection rates from the other carcinogenic HPV types that are responsible for the remaining 10% of cervical cancer. Ideally, an even broader vaccine that could protect against these other types would help to avoid such concerns. However, the high dosage already required for Gardasil 9 (270 μg protein in a single dose) presents a significant challenge for the development of such a broad-acting vaccine through standard manufacturing practices.

Previous studies on HPV structures have shown that the HPV L1 monomer forms a canonical, eight-stranded β-barrel (BIDG-CHEF), and both biochemical[22–27] and structural analyses[28–30] indicate that HPV type-specificity is largely dependent on five highly variable surface loops—BC, DE, EF, FG and HI—on the HPV capsid[31,32]. It stands to reason that a chimeric recombinant L1 capsid comprising the antigenic determinants of different HPV types may present an alternative strategy for the development of a cross-type or wide-spectrum HPV vaccine. Previous reports have observed that the recognition region (i.e., homologous sequences on the surface loops) of type-specific neutralizing antibodies of one HPV type (type A) could be transferred to create a chimeric VLP of another type (type B)[23,33–35]. However, previous studies show that such a chimeric VLP of type B only generated low levels of neutralizing antibodies that could cross-react with type A[22,34], and because of this poor response, there was little further interest in developing chimeric L1 VLPs based on such epitope grafting.

In this study, we sought to develop a highly efficacious, cross-type vaccine candidate against the infection of two or more HPV types using an epitope grafting approach. We initially found that the L1 proteins of various HPV types shared an overall conserved structure in their core regions and even in their basic surface loop configurations, with phenotype variations between different HPV types able to be induced by minor structural movements of the surface loops. Our design is based on a rational understanding of the high structural conservation of L1 surface loops within phylogenetically close HPV types despite sequence variation; and we used this knowledge to select among hundreds of HPV variants to engineer chimeric particles. Here we characterized a chimeric HPV cross-type vaccine candidate with comparable efficacy to that of mixed wild-type (WT) VLPs. Our study confirms that the strict type specificity of HPV can be undermined by the introduction of a few residue substitutions between genetically close HPV types, introducing an appealing strategy for the design and development of wider-spectrum vaccines against HPV as well as other pathogenic microbes.

## Results

**Genetically close HPV types share high structural similarity.** The type-specific neutralization epitopes of HPV, which are located on the surface of the HPV capsid, are mostly mediated by the pentamers[36]. To ascertain whether structural divergence drives HPV type specificity, we scrutinized the crystal structures of L1 pentamers of genetically close or distant HPV types (Fig. 1a, b). First, L1 proteins of HPV33 and HPV52 were cloned and expressed in *E.coli*, purified, digested by trypsin, and crystalized. The crystal structures of HPV33 and HPV52 pentamers were determined to resolutions of 2.9Å and 2.75Å, respectively (Table 1, Supplementary Fig. 1a), with refinement to $R_{work}/R_{free}$ values of 21.0/24.0% and 18.7/23.5%, respectively. Both structures show a hollow lumen in the central axis, and two neighboring monomers intertwined by surface loops (Fig. 1b, Supplementary Fig. 1a). Each L1 monomer folds into a canonical, eight-stranded β-barrel (BIDG-CHEF), with the strands joined via flexible surface loops (BC, DE, EF, FG, HI; Supplementary Fig. 1b, c). The same secondary structural elemental profile is found among various L1 structures (HPV11/16/18/35/58/59)[12,31].

Next, we sought to determine the extent of the differences in the tertiary structures of L1 proteins among different HPV types; these are well defined in the electron density map, even for the variable loops located on the outer surface (Supplementary Fig. 2). It should be noted that the surface loops in the non-crystallographic symmetry (NCS)-related monomers within the asymmetric unit of HPV33, HPV52, and HPV58 might differ

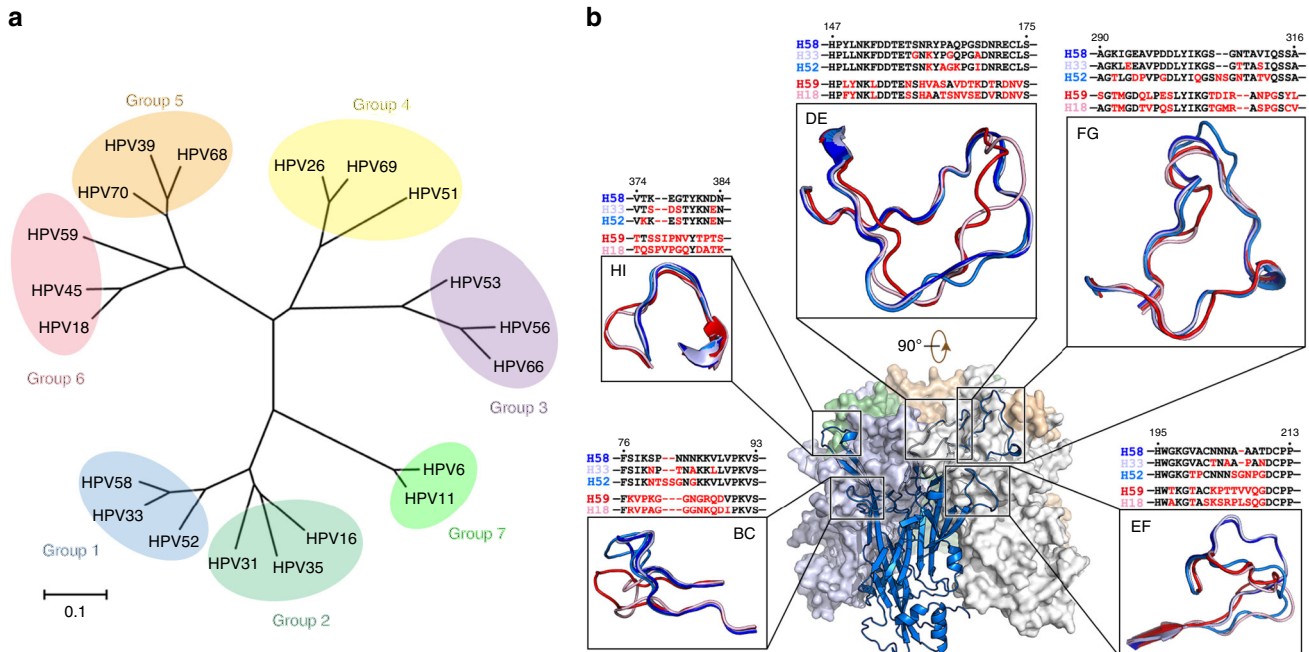

**Fig. 1** Phylogenetic classification among the major capsid proteins (L1) of multiple types, and a structural comparison of HPV L1s. **a** Phylogenetic tree of HPV L1 proteins based on the amino acid sequences of 20 HPV types. The bar at the bottom provides a scale for the change in evolutionary lineages over time. Twenty HPV types were divided into seven groups based on evolutionary distance of their L1 proteins: Group 1 (blue): HPV33, -52 and -58; Group 2 (cyan): HPV16, -31 and -35; Group 3 (purple): HPV53, -56 and -66; Group 4 (yellow): HPV26, -51 and -69; Group 5 (orange): HPV39, -68 and -70; Group 6 (pink): HPV18, -45 and -59; and Group 7 (green): HPV6 and -11. **b** Structural comparison of loop structure among HPV types of HPV33, -58 (PDB code: 5Y9E), -52, -18 (PDB code: 2R5I), and -59 (PDB code: 5J6R). The loops are colored by type: HPV33 (light blue), HPV58 (deep blue), HPV52 (marine), HPV18 (light pink) and HPV59 (red). The loop sequence alignments among types are shown above the corresponding loop structure, with residue differences compared with HPV58 colored red

### Table 1 Data collection and refinement statistics

| | HPV33 | HPV52 | H58-33BC | H58-33BC-52HI |
|---|---|---|---|---|
| Data collection | | | | |
| Space group | P2$_1$ | C2 | P2$_1$ | P1 |
| Cell dimensions | | | | |
| $a, b, c$ (Å) | 98.8, 171.9, 145.7 | 306.8, 105.1, 196.9 | 153.7, 105.8, 154.7 | 136.5, 209.8, 212.6 |
| $\alpha, \beta, \gamma$ (°) | 90.0, 97.0, 90.0 | 90.0, 125.8, 90.0 | 90.0, 99.5, 90.0 | 60.5, 85.1, 90.1 |
| Resolution (Å) | 50.0–2.9 (2.97–2.92)[a,b] | 50.0–2.7 (2.80–2.75) | 50.0–2.5 (2.54–2.50) | 50.0–3.5 (3.56–3.50) |
| $R_{sym}$ (%) | 9.5 (75.1) | 19.9 (123.8) | 13.4 (83.3) | 17.1 (64.5) |
| $I / \sigma I$ | 14.2 (2.1) | 12.1 (1.8) | 11.1 (1.8) | 4.8 (1.3) |
| Completeness (%) | 99.8 (99.9) | 99.9 (100.0) | 99.8 (99.7) | 97.0 (87.2) |
| Redundancy | 3.7 (3.7) | 7.6 (7.6) | 3.7 (3.7) | 1.9 (1.8) |
| Refinement | | | | |
| Resolution (Å) | 48.2–2.9 | 42.7–2.8 | 40.8–2.5 | 40.4–3.5 |
| No. reflections | 105,166 | 131,736 | 169,776 | 248,099 |
| $R_{work} / R_{free}$ | 21.0/24.0 | 18.7/23.5 | 18.2/21.6 | 31.5/34.0 |
| No. atoms | 33,033 | 33,256 | 34,661 | 133,168 |
| Protein | 33,033 | 33,256 | 33,241 | 133,168 |
| Water | — | — | 1420 | — |
| $B$-factors | 76.1 | 53.0 | 40.7 | 105.5 |
| Protein | 76.1 | 53.0 | 40.9 | 105.5 |
| Water | — | — | 35.0 | — |
| R.m.s. deviations | | | | |
| Bond lengths (Å) | 0.003 | 0.007 | 0.004 | 0.003 |
| Bond angles (°) | 0.65 | 0.93 | 0.67 | 0.62 |

[a]One crystal was used for each structure.
[b]Values in parentheses are for highest-resolution shell

slightly in conformation due to their intrinsic flexibilities (Supplementary Fig. 3). First, we divided 20 of the major HPV types into seven groups based on evolutionary distance of L1 proteins: 18 of these are responsible for more than 99% of cervical cancer, with HPV6 and 11, in particular, accountable for about 90% of genital warts (Fig. 1a, Supplementary Table 1). Then, five L1 pentamer structures representing two groups (group 1: HPV58, -33 and -52; group 6: HPV18 and -59) were aligned to compare the structural differences among their five surface loops. We show that the overall loop structures are conserved for all five surface loops among various types of L1 structures. Nevertheless, the variable sequence identity of individual genotypes induces small loop movements between types, as shown in the superimposed structures. We also observe that the surface loops of types within group 1 or group 6 have quite similar main-chain traces, with minor differences noted among the loops between the two groups. The extent of loop movement between L1s are correlated with aa sequence diversity in terms of residue type and sequence length (Fig. 1b). Furthermore, pairwise structural comparisons were carried out between NCS copies of L1 monomers from five HPV types and the mean root mean squared deviations (RMSDs) were generated for both the core regions and the surface loops of L1 for the various genotypes (Supplementary Table 2). In the comprehensive structural comparisons, both the core region structures and the surface variable regions—constituted by five surface loops among the various types of L1 structures—shared conserved structural configurations, with RMSD values of Cα ranging from 0.22 to 0.59Å and from 0.53 to 2.06Å, respectively (Supplementary Table 2). The minor loop movements, particularly between types from group 1 and group 6 (Fig. 1b), slightly increased the RMSD values with respect to that of the relatively rigid core region, which was believed to be associated with the various type-specific phenotype variations underlying virus phylogenetic evolution. Collectively, the structural information suggests high conservation of the overall structures of L1 within a phylogenetic group of HPV, which even extends across different groups, despite minor variations in the outer antigenic surface loops.

**Design and expression of chimeric HPV VLPs.** Viral epitopes that induce neutralizing antibodies against HPV seem to cover more than one surface loop on a single L1 protein[25,28,29,37,38]; this may explain why high variability in the loop region leads to type specificity among different HPV genotypes. Therefore, our analysis on the structure of HPV33/58/52/18/59 types demonstrated that the L1 surface loops showed high structural similarities within a phylogenetic group, which remarkably extended across different groups, with only minor variation in the antigenic loops. This type of conserved structure despite genotypic evolution illustrates the variability that can be induced while conserving function, and then provides preliminary structural information to explain the cross-protection achieved by HPV vaccines against infection with a limited number of phylogenetically related HPV types[39,40]. We thus propose that highly potent cross-type protection against HPV infection could be achieved by creating a chimeric HPV VLP that contains antigenic determinants from genetically close HPV types in same phylogenetic group rather than evolutionarily distant HPV types across groups. Given the extent of the structural diversity among the five surface loops, it is assumed that comprehensive loop swapping trials could be implemented to screen for an appropriate chimera that may confer the expected cross-type antigenicity and immunogenicity.

To test our hypothesis, we genetically swapped the loop sequences on five surface regions (BC/DE/EF/FG/HI) between the two most phylogenetically related HPV types—HPV33 and HPV58—and generated 10 chimeric L1 constructs: H33-58BC, H33-58DE, H33-58EF, H33-58FG, H33-58HI, H58-33BC, H58-33DE, H58-33EF, H58-33FG and H58-33HI (Fig. 2a and Supplementary Fig. 4). These mutants were expressed in *E.coli* (Supplementary Fig. 5a, b), and self-assembly of the purified mutated L1 proteins was confirmed by transmission electron

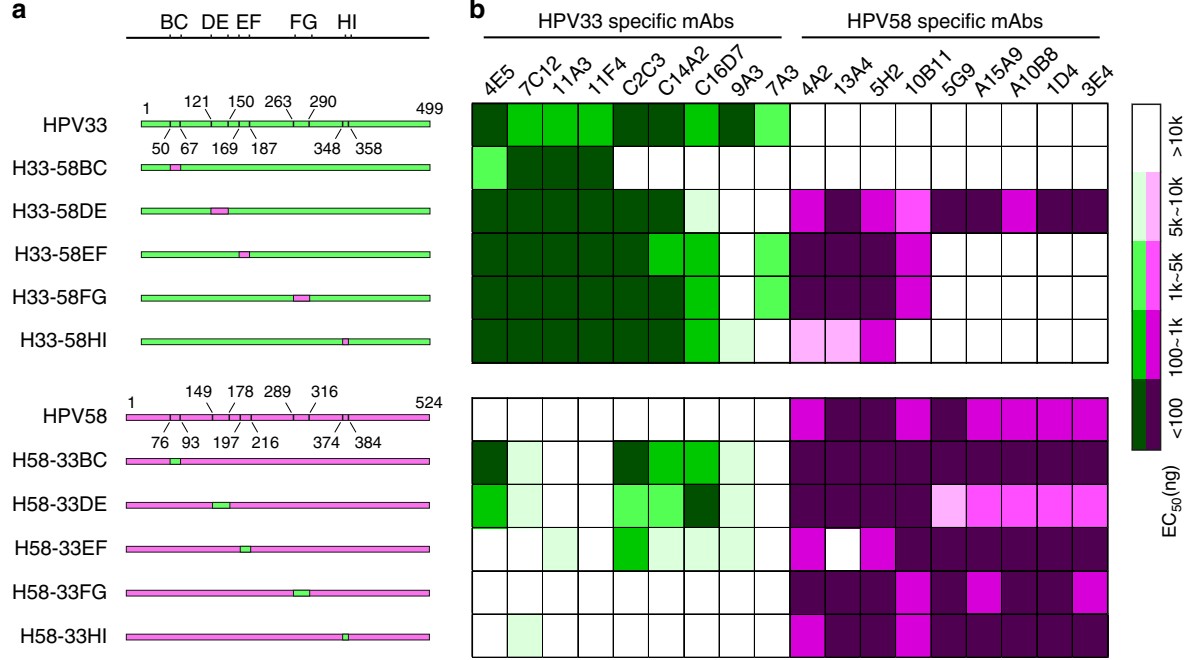

**Fig. 2** Molecular design and antigenic characterization of loop-swapped HPV33/58 chimeric VLPs. **a** Schematic representation of the wild types (WTs) and the chimeric HPV33/58 L1 proteins. The location of surface loops for each WT or mutated L1 protein are labeled by amino acid position. **b** Heatmap representations of the EC$_{50}$ values of chimeric HPV33/58 VLPs based on ELISA assays against a type-specific mAb panel of HPV33 and HPV58 VLPs. The key indicates the heatmap gradient. A detailed characterization of each mAb is shown in Supplementary Table 3

microscopy (Supplementary Fig. 5c). Particle size and homogeneity of the chimeric HPV VLPs, as determined by high-performance size-exclusion chromatography (HPSEC) and analytical ultra-centrifugation (AUC), were similar to those measures observed for WT L1 VLPs of their corresponding backbone type (Supplementary Fig. 5c). Using differential scanning calorimetry (DSC), we found that the transition corresponding to the melting temperature (Tm) in the DSC trace occurred at 77.28 °C for WT HPV33 VLPs and 68.61 °C for WT HPV58 VLPs, whereas the Tm values for the chimeric HPV33 VLPs (H33-58BC, H33-58DE, H33-58EF, H33-58FG and H33-58HI) and chimeric HPV58 VLPs (H58-33BC, H58-33DE, H58-33EF, H58-33FG and H58-33HI) ranged from 61 to 74 °C and 62 to 67 °C, respectively. These results indicated no obvious effects on the structural stabilities of the L1 VLPs following the amino acids substitutions on the capsid surface regions (Supplementary Fig. 5).

**Antigenicity and immunogenicity of chimeric HPV33/58 VLPs**. To investigate the antigenicity of the chimeric HPV33/58 VLPs, we employed a panel of neutralizing antibodies against HPV33 and HPV58 (Supplementary Table 3); this allowed us to evaluate the binding ability of the VLP mutants to those type-specific antibodies. The median effective concentrations ($EC_{50}$) of the mAb panel to various chimeric HPV33/58 VLPs was determined by ELISA assay (Fig. 2b). We observed the following three main findings: (1) The binding of HPV33 (or HPV58) mAbs was affected, to varying degrees, by loop substitutions at one of the five surface loops (Fig. 2b), suggesting that particular loop(s) are responsible for mAb recognition. Notably, five of the nine HPV33-specific mAbs (C2C3, C14A2, C16D7, 9A3 and 7A3) failed to bind to the BC loop-swapped mutant of HPV33 (H33-58BC), suggesting that the BC loop of HPV33 may involve a major neutralization epitope. (2) Enhanced binding activity of multiple non-backbone-type mAbs to chimeric VLPs was found following the substitution of several amino acids on different surface loops of HPV58 into HPV33 (and vice versa). Specially, eight HPV58-specific mAbs showed binding to the DE loop-swapped mutant of HPV33 (H33-58DE; Fig. 2b and Supplementary Table. 3). (3) Binding of several mAbs to WT VLPs could be disrupted by single loop substitutions with the heterologous type; i.e., replacement of one of the four loops (BC, DE, EF and FG) in WT HPV33 VLP abrogated the binding of the specific mAb 9A3, suggesting that these four loops are involved in mAb 9A3 recognition of HPV33 VLP.

To assess the immunogenicity of the cross-type chimeric HPV33/58 VLPs, BALB/c mice were inoculated three times with chimeric VLPs formulated with aluminum adjuvant. The neutralization titers of the sera were measured with a pseudo-virion–based cell neutralization assay. We noticed that the WT HPV33 and HPV58 VLPs can elicit moderate reciprocal cross-type neutralization titers ranging from $10^{1.2}$ to $10^{2.6}$, which is consistent with previous results[41]. Meanwhile, all of the chimeric HPV33/58 VLPs showed improved heterotypic antibody responses, and this occurred without changing the elicitation of neutralization antibodies against their own backbone types (Fig. 3a). Notably, the anti-HPV58 antibody titers of H33-58DE and H33-58HI were 25-fold (titers ranging from 10,000 to 102,400; $P < 0.01$) and 80-fold (102,400 to 212,400; $P < 0.001$) higher than that found in mice immunized with WT HPV33 VLPs (2.5 to 102,400), respectively (Fig. 3a), whereas H58-33BC, H58-33DE, and H58-33HI could induce, respectively, 325-fold (24,000 to 204,800; $P < 0.001$), 94-fold (2,000 to 51,200; $P < 0.01$), and 52-fold (1,700 to 33,000; $P < 0.01$) higher titers against PsV33 than that elicited by WT HPV58 VLPs (Fig. 3a).

H33-58HI and H58-33BC mutants showed comparable cross-protection potency as that conferred by bivalent HPV33 and HPV58 VLPs against both HPV33 and HPV58 (Fig. 3a). Therefore, these two mutants were subjected to further study, testing immunogenicity assays in mice. We found that immunization with higher concentrations per dose led to higher titers: animals receiving 0.1 μg per dose of chimeric VLP H33-58HI and H58-33BC were capable of inducing anti-HPV58 and anti-HPV33 neutralization titers of up to 49,500 and 29,000, respectively (Fig. 3b); 1 μg per dose led to 1.5-fold (50,000–92,000) and 3.3-fold (64,000–126,000) higher neutralization titers, respectively, than with 0.1 μg per dose; and 10 μg per dose led to 2.0-fold (102,000–192,000) and 1.5-fold (100,000–195,000) higher neutralization titers, respectively, than with 0.1 μg per dose. These results were reflected by the $ED_{50}$ values of the chimeric VLPs; the $ED_{50}$ values define the minimum effective dose that provides seroconversion in 50% of the animals in a given group[42]. As shown in Table 2 and Supplementary Table 4, the $ED_{50}$ values of H33-58HI were as low as 0.053 μg for HPV33 and 0.049 μg for HPV58, whereas, for H58-33BC, the values were 0.043 μg and 0.019 μg, respectively. This indicates that VLPs of both H33-58HI and H58-33BC could provide cross protection against HPV33 and HPV58 at levels comparable with that of WT HPV58 and HPV33, respectively (Table 2).

Furthermore, we questioned whether the cross protection afforded by these two chimeras was achieved by multiple simultaneous mutations. We generated eight single mutation chimeras of HPV33 VLPs (mutated to the HPV58 HI loop: H33-S350K, H33-D351E, H33-S352G, H33-E357D) and HPV58 VLPs (mutated to the HPV33 BC loop: H58-S80N, H58-N82T, H58-N84A, H58-V87L). The neutralization titers of the anti-sera of these chimeras were measured and we found that most of the single-point mutants (except for H58-V87L) could elicit heterotypic neutralization antibodies (Supplementary Fig. 6) to some extent; this indicated that every residue associated with type-restricted neutralization contributes to the cross-neutralization when substituted to the other type. Intriguingly, the H33-E357D chimera was able to induce a cross-type neutralization titer as high as that of H33-58HI (Supplementary Fig. 6), possibly demonstrating that residue D377 of HPV58 L1 takes a critical role during the immune recognition process of HPV58.

To check if cross-protection could be achieved by introducing mutations from more distantly related HPV types, we generated two chimeric VLPs—H33-59HI and H58-59BC—using the distant type HPV59, which belongs to another group in the phylogenetic tree (Fig. 1a; Supplementary Fig. 7a, b). To evaluate their potential cross-type immunogenicity, mice were immunized three times with a higher dose (100 μg) of proteins absorbed with a stronger Freund's adjuvant (Supplementary Fig. 7c). In contrast to the high neutralization titers against heterologous types elicited by H33-58HI and H58-33BC, we found no increase in the anti-HPV59 antibody response in the antisera of either H33-59HI or H58-59BC (Supplementary Fig. 7c). Thus, cross-protective immunities was not be elicited by proteins engineered using genetically distant types.

**Immunogenicity of triple-type HPV33/58/52 chimeras**. To explore the feasibility of including more genetically close genotypes into a single, chimeric L1 VLP type, HPV52—which is closest to both HPV33 and HPV58 (Fig. 1a)—was engineered into the lead HPV33/58 chimeric VLPs, H33-58HI and H58-33BC, by individually swapping each of the remaining four loops with those from HPV52. This resulted in eight HPV33/58/52 chimeras: H33-58HI-52BC, H33-58HI-52DE, H33-58HI-52EF, H33-58HI-52FG, H58-33BC-52DE, H58-33BC-52EF, H58-33BC-52FG and H58-33BC-52HI (Fig. 4 and Supplementary Fig. 8a–c). The VLPs from these chimeras were then used to immunize mice

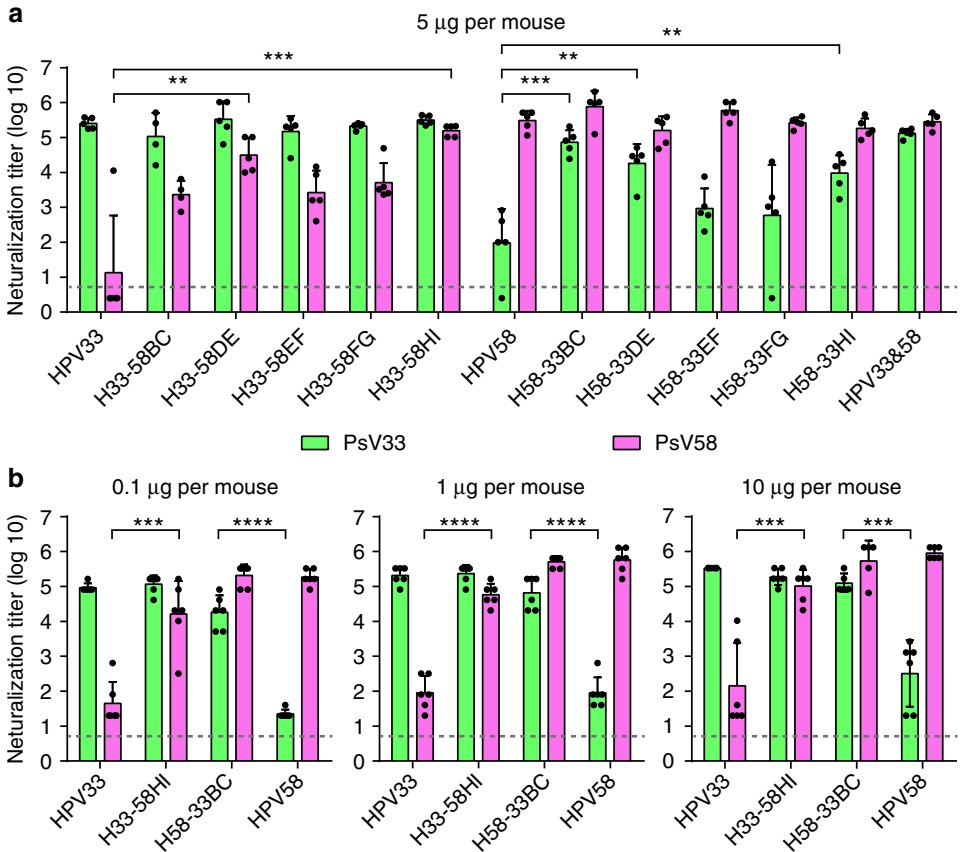

**Fig. 3** Immunogenicity of loop-swapped HPV33/58 chimeric VLPs. **a** Neutralization titers of immune sera from BALB/c mice ($n = 5$) collected four weeks after the third immunization (immunization at weeks 0, 2 and 4 with 5 μg per dose of aluminum adjuvant-containing chimeric or WT VLPs). A group of mice immunized with mixed WT HPV33 and HPV58 VLPs was labeled. (*) denotes significant differences as compared with the neutralization titer induced by the corresponding backbone type. **b** Groups of mice ($n = 6$) were immunized with 0.1, 1 or 10 μg per dose of the selected chimeric or WT HPV33/58 VLPs at weeks 0, 2 and 4. Immune sera were collected at week 8 and analyzed by neutralization assays. All the data were analyzed by one-way analysis of variance (ANOVA), **$P < 0.01$; ***$P < 0.001$; ****$P < 0.0001$. The error bars are standard deviation and symbols represent individual mice. The dotted line indicates the limit of detection for the assay. Non-neutralizing sera were assigned a value of one-half of the limit of detection for visualization

## Table 2 Half-effective dosage (ED50) of double-type and triple-type chimeras in mice

| Antigen category | HPV VLP | ED50 (μg) | | |
| --- | --- | --- | --- | --- |
| | | anti-HPV33 | anti-HPV58 | anti-HPV52 |
| Wild-type control | HPV33 | 0.066[a] | > 0.3 | — |
| | HPV58 | > 0.3 | 0.006 | — |
| Double-type chimera | H33-58HI | 0.053 | 0.049 | — |
| | H58-33BC | 0.043 | 0.019 | — |
| Wild-type control | HPV33 | 0.013[b] | >0.3 | >0.3 |
| | HPV58 | > 0.3 | 0.021 | >0.3 |
| | HPV52 | >0.3 | >0.3 | 0.046 |
| | H33&58&52 | 0.009 | 0.009 | 0.017 |
| Triple-type chimera | H33-58HI-52DE | 0.083 | 0.069 | 0.557 |
| | H33-58HI-52FG | 0.046 | 0.243 | 0.087 |
| | H58-33BC-52DE | 0.009 | 0.014 | 0.341 |
| | H58-33BC-52HI | 0.065 | 0.058 | 0.041 |

[a]Refer to the detailed ED50 calculation in Supplementary Table 4
[b]Refer to the detailed ED50 calculation in Supplementary Table 5

to assess their cross-type immunogenicity. Of the eight triple-type chimeric VLPs, H33-58HI-52DE and H33-58HI-52FG showed the highest anti-HPV52 antibody titers (4700–2200 and 1300–47900, respectively; $P < 0.001$) as compared with the H33-58HI control. H58-33BC-52DE and H58-33BC-52HI also showed high titers (2,280–23,440 and 2,180–87,430, respectively; $P < 0.001$) as compared with the H58-33BC control (Fig. 4b). No anti-HPV52 neutralizing antibodies were found for any of the controls (WT HPV33, WT HPV58, or HPV33/58 chimeras). Intriguingly, two of the triple chimeras—H33-58HI-52FG and H58-33BC-52FG—showed lower anti-HPV58 neutralizing titers than those of their backbone VLPs (30-fold [1,700–10,640] and 330-fold [2.5–5,400] for H33-58HI and H58-33BC, respectively); this suggested that the FG loop (in HPV58 in particular) was part of the immunodominant region associated with the neutralizing antibody production. Anti-HPV33 neutralization titers for all triple chimeras remained comparable with that of their parental double chimeras.

Next, we further measured the anti-HPV52 immunogenicity of four selected candidates—H33-58HI-52DE, H33-58HI-52FG, H58-33BC-52DE and H58-33BC-52HI—using three different immunization doses (0.1, 1 and 10 μg), as before. All four mutants showed dose-dependent anti-HPV52 neutralization antibody responses (Fig. 5a). However, H33-58HI-52FG, which showed moderate antibody titers (2,280–23,440) in the cross-type immunogenicity assays, failed to show high potency against

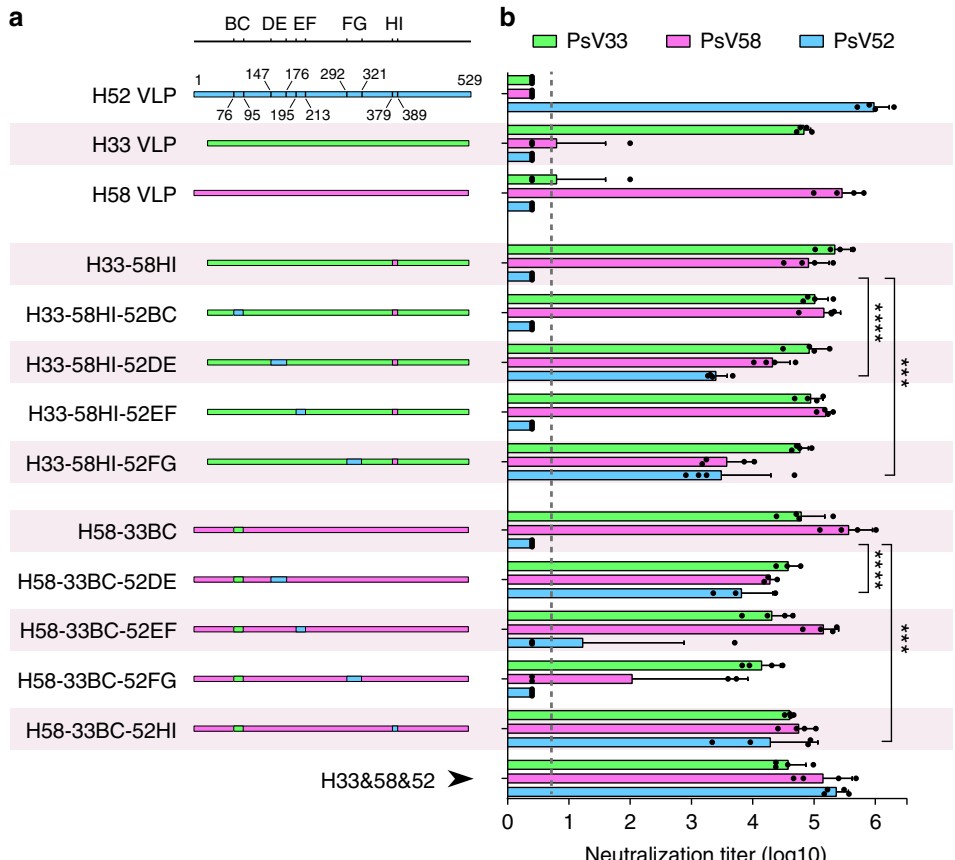

**Fig. 4** Immunogenicity of loop-swapped HPV33/58/52 chimeric VLPs. **a** Schematic representation of the chimeric HPV33/52/58 L1 proteins based on chimeric L1 proteins H33-58HI and H58-33BC. A group of mice were immunized with mixed WT HPV33, -52 and -58 VLPs. **b** Neutralization titers of the immune sera from BALB/c mice ($n = 4$) collected 4 weeks after the third immunization (immunization at weeks 0, 2 and 4 with 5 μg per dose of aluminum adjuvant-containing tripe-type chimeric or WT VLPs). All the data were analyzed by one-way ANOVA, **$P < 0.01$; ***$P < 0.001$; ****$P < 0.0001$. The error bars are standard deviation and symbols represent individual mice. The dotted line indicates the limit of detection for the assay. Non-neutralizing sera were assigned a value of one-half of the limit of detection for visualization

HPV52 infection in groups receiving doses as low as 0.1 μg or 1 μg (Fig. 5a, left and middle). Also, the $ED_{50}$ values for these four chimeras were measured to test their immunogenicity against infection of all three HPV types. Overall, only H58-33BC-52HI conferred high immune efficacy against infections from all three genotypes (HPV33 [$ED_{50}$, 0.065 μg], HPV52 [0.041 μg] and HPV58 [0.058 μg]), reaching comparable levels to those provided by a mix of the WT VLPs of HPV33, -52, and -58 (0.009 μg, 0.017 μg, and 0.009 μg, respectively; Table 2 and Supplementary Table 5).

**Immunogenicity of other chimeric VLP combinations.** To test whether this method could be applied to produce other triple HPV types, we attempted to graft immunodominant epitopes into a single L1 molecule within different groups. Four chimeras (H35-16FG-31BC, H66-56HI-53FG, H39-68FG-70HI and H45-18BC-59HI) were generated for Groups 2, -3, -5 and -6 (Fig. 1a) based on the protocol summarized for the generation of HPV58-33BC-52HI, and were tested in mice (5 μg per dose). All four chimeric VLPs showed high cross-protective efficacy for each group (Fig. 5b and Supplementary Fig. 9), with at least 150-fold increased heterotypic neutralization titer achieved by these lead candidates as compared with the backbone type, without significantly affecting the original immunization response (Supplementary Table 6). Thus, we believe that engineering in the surface loops between closely related HPV types can be used to optimally modulate the immunodominance among different types.

**Immunogenicity and immune durability in non-human primates.** We next immunized cynomolgus macaques (*Macaca Fascicularis*) with the lead HPV33/58/52 vaccine candidate to evaluate vaccine-induced immunity. Monkeys ($n = 4$ per group) were administered with 5 μg of the H58-33BC-52HI chimeric VLPs formulated with aluminum adjuvant at week 0 and 8. Neutralization titers were measured from antisera drawn at week 10. Immunization with WT HPV VLPs resulted in minor cross-neutralization of HPV33 and HPV58 in monkeys (titers ranging from 50 to 500); however, no cross-reactive antibodies to HPV52 were observed with HPV33 or HPV58 WT VLPs, and similarly, no cross-reactive antibodies to HPV33 or HPV58 were detectable after HPV52 VLP immunization (Fig. 6a). Nevertheless, the cross-reactive antibody titers for HPV33 were significantly (26-fold) higher with the chimeric VLPs (960 to 6,290, $P < 0.001$; Fig. 4a, left) than with the HPV58 WT VLPs; reciprocally, the anti-HPV58 antibody titer (6,060 to 40,960) of H58-33BC-52HI was also 54-fold higher than that of HPV33 WT VLPs ($P < 0.001$; Fig. 6a, right). The chimeric VLPs could also elicit a mean neutralization titer of 5,400 (1,780 to 8,460) against HPV52 (Fig. 6a, middle). Thus, engineered loop swapping can produce high immune potency and provide cross-type protection of phylogenetically related HPV types.

We then tested the kinetics of the vaccine-induced neutralizing antibodies in immunized monkeys. Two weeks after the first vaccination, the anti-HPV33 neutralizing titer reached ~200, with the titers then slightly decreasing until the second immunization

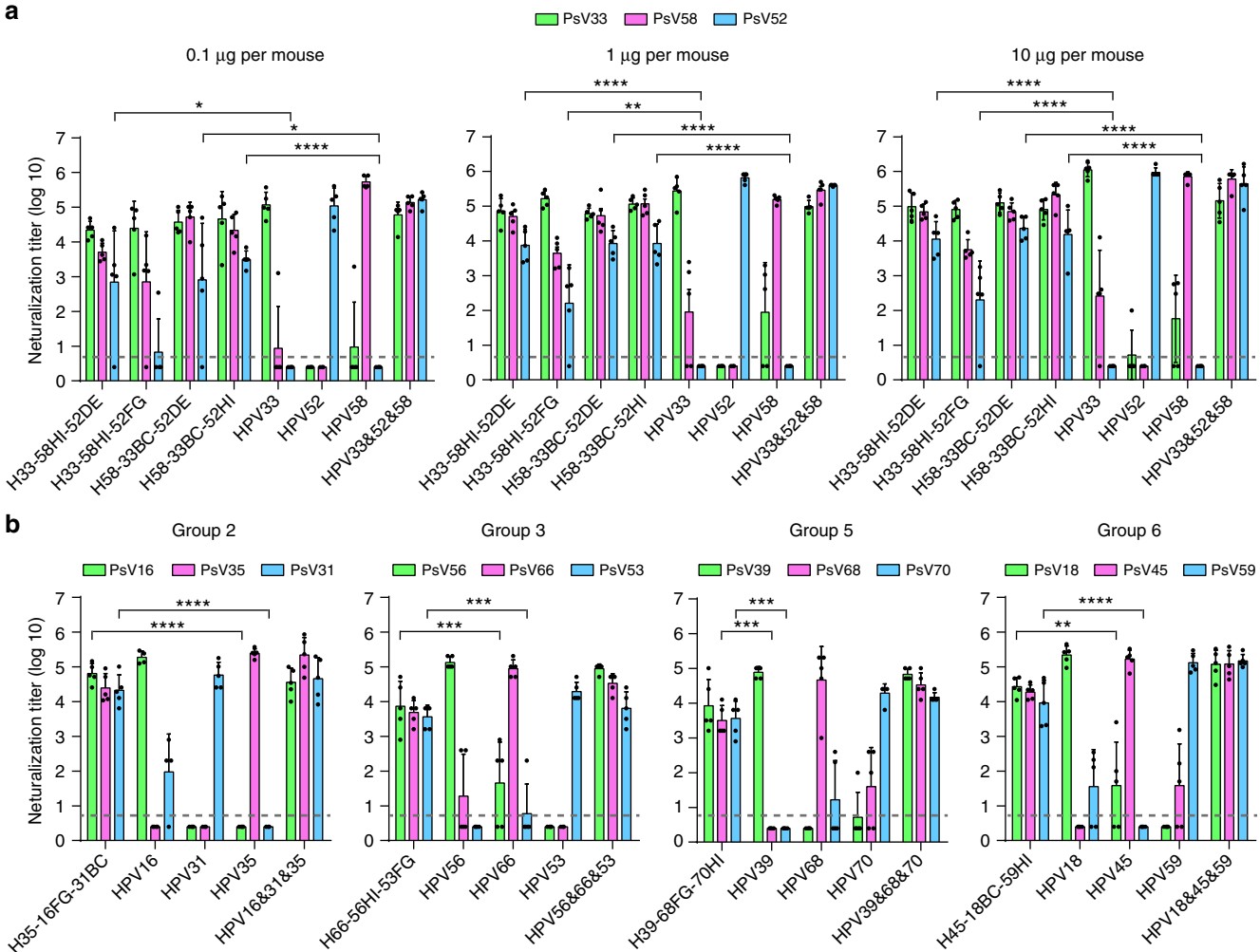

**Fig. 5** Immunogenicity of loop-swapped HPV33/58/52 chimeric VLPs at three dosages and other triple-type chimeric VLPs. **a** Groups of mice ($n = 5$) were immunized with 0.1, 1 or 10 μg per dose of selected chimeric or WT VLPs at weeks 0, 2 and 4. Immune sera were collected at week 8 and analyzed by neutralization assays. **b** Neutralization titers of immune sera (at week 8) in BALB/c mice ($n = 5$) immunized with 5 μg per dose of chimeric and WT L1 proteins of Groups 2, -3, -5, and -6 (also see Fig. 1a) in aluminum adjuvant at weeks 0, 2 and 4. All the data were analyzed by one-way ANOVA, **$P < 0.01$; ***$P < 0.001$; ****$P < 0.0001$. The error bars are standard deviation and symbols represent individual mice. The dotted line indicates the limit of detection for the assay. Non-neutralizing sera were assigned a value of one-half of the limit of detection for visualization

at week 8 (Fig. 6b, left). Titer levels peaked at four weeks after the second vaccination (week 12), slightly decreasing over time but remaining high even after four and a half months (~2,200 at week 26); albeit, this titer was ~2-fold lower than that of mixed WT VLPs of HPV33/52/58 (Fig. 6b, left). Similar kinetic curves were observed for anti-HPV52 (Fig. 6b, middle) and anti-HPV58 neutralizing titers (Fig. 6b, right), with the overall titer against HPV52 and HPV58 reaching 1,100 and 22,000 at week 26, respectively (Fig. 6b). Neutralizing antibody titers in monkeys immunized with WT HPV33 or WT HPV58 VLPs showed increased reciprocal neutralization after the first and second vaccinations but these also reduced over time, with low cross-protective antibody titers of ~25 (anti-HPV58) for HPV33 and ~33 (anti-HPV33) for HPV58 by week 26 (Fig. 6b, left and right). In monkeys receiving HPV52 VLPs, no cross-reactive antibodies were found for either HPV33 or HPV58 at week 26 (Fig. 6b, left and right), as anticipated. Overall, the data confirm the high immune potency and cross-type protection afforded by the HPV33/52/58 chimeric VLPs designed through the exchange of several amino acids between phylogenetically close related types (Fig. 7a and Supplementary Fig. 10), and also indicate that

antibody production can be enhanced in pre-existing immunity and immunization with these chimeric VLPs to provide long-lasting (~4.5 months) protection, at levels comparable with that achieved by mixing three WT HPV VLPs at three times the dosage (versus one dose with the chimeric VLPs; Fig. 6b).

**Structures of chimeric pentamers and VLPs-Fab complexes.** We solved the crystal structures of pentamers of the lead double and triple chimeric candidates H58-33BC and H58-33BC-52HI to resolutions of 2.5Å and 3.5Å, respectively (Fig. 7b and Supplementary Fig. 11a; Table 1). The two pentamer structures of the chimeric mutants were superimposed against wild-type structures of HPV58, 33 and 52 to identify marked structural differences (Fig. 7b-d). Consistent with the high structural similarities within this phylogenetic group, as indicated in Fig. 1, the type-specific aa replacement in both the BC loop and HI loop still did not substantially change the main-chain conformation of these two loops in the chimeras (Fig 7c, d top left). Further pairwise comparisons of the surface contours of wild types and mutants illustrated that the most distinct bumpy protrusions between parental and

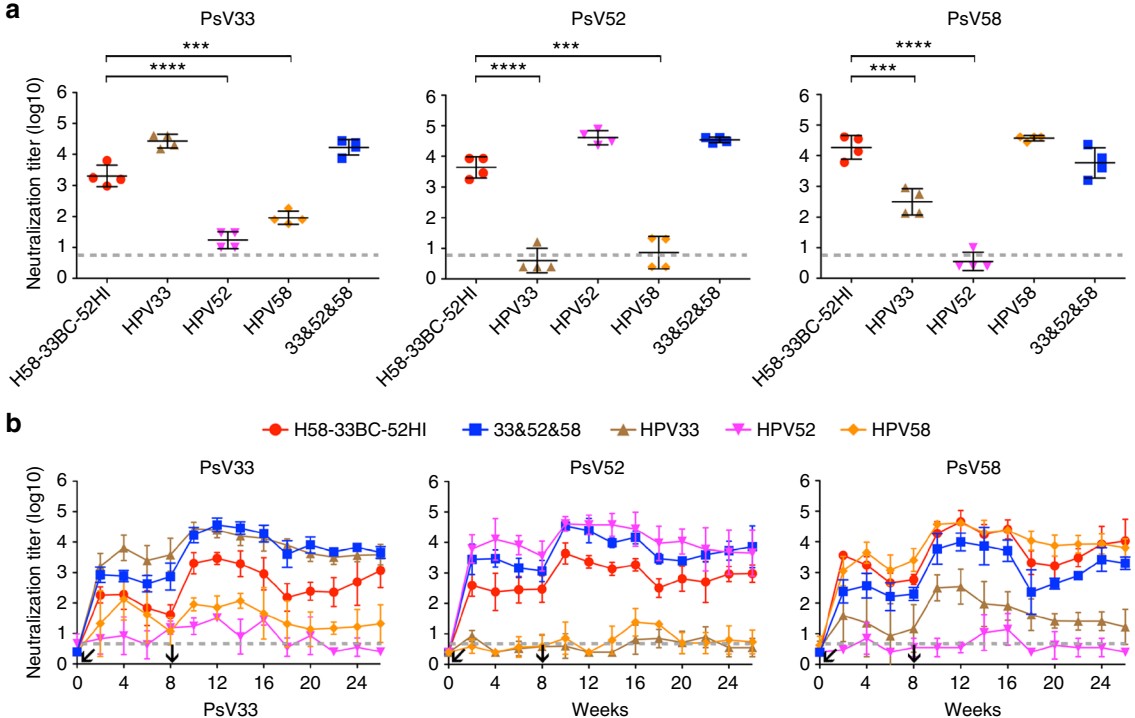

**Fig. 6** Immunogenicity of triple-type chimeric HPV33/52/58 VLPs in Cynomolgus Macaques. **a** Neutralization titers against PsV33, PsV52 and PsV58, respectively, from the immune sera in monkeys ($n = 4$) collected two weeks after the second immunization (immunization at weeks 0 and 8 with 5 μg per dose of aluminum adjuvant-containing chimeric H58-33BC-52HI or WT VLPs). Each symbol represents one monkey. Statistical significance was assessed by one-way ANOVA, **$P < 0.01$; ***$P < 0.001$; ****$P < 0.0001$. **b** Kinetics of serum neutralization titers in monkeys after immunization with the chimeric or WT VLPs. Immune sera were collected weekly after immunization, and serum neutralization titers were determined and plotted. The dotted line indicates the limit of detection for the assay. Non-neutralizing sera were assigned a value of half of the limit of detection for visualization. All the neutralizing titers were calculated to express the mean value with SD

targeting types were located on the side-chains of $K53^{H33}$, $N57^{H33}$, $K59^{H33}$ and $K60^{H33}$ for HPV58 vs. HPV33 (Fig. 7c, bottom left), and $K380^{H52}$, $S383^{H52}$ and $E388^{H52}$ for HPV58 vs. HPV52 (Fig. 7d, bottom left). By loop swapping with a few aa mutations on HPV58 L1—i.e., S80N, N82T, N84A and V87L for BC loop, T375K, G378S and D383E for HI loop (Fig. 7a)—the resurfaced structural contours of the BC and HI loops of the chimera nearly recapitulate those of the HPV33 BC loop (Fig. 7c, middle and right) and HPV52 HI loop (Fig. 7d, right), respectively. These subtle structural resurfacing changes in the engineered regions might provide structural evidence for the cross-protective immunity of the chimeric proteins.

To verify that the residues exchanged on the surface loops of HPV58 introduce a similar functional surface area as that of the corresponding heterologous type, we recruited mAb 4E5—a conformational and type-specific neutralizing antibody of HPV33, the binding of which could be redirected to H58-33BC and H52-33BC-52HI (Figs 2b, 8a, c, d, Supplementary Fig. 12 and Supplementary Table 7)—and determined cryoEM structures of the 4E5 Fab in complex with HPV33 WT VLPs (HPV33:4E5) and both chimeric VLPs (H58-33BC:4E5 and H58-33BC-52HI:4E5). The cryo-EM structures of HPV33:4E5, H58-33BC:4E5 and H58-33BC-52HI:4E5 were reconstructed to 12.3Å, 12.0Å and 10.0Å, respectively (Fig. 8d, Supplementary Fig. 13; Supplementary Table 8). From the cryoEM micrographs and 2D class averages, we can discern obvious protrusions, which we consider as the bound 4E5 Fabs, for both chimeric VLPs:4E5 complexes, similar to that observed for the HPV33:4E5 complex (Supplementary Figs. 13a, 14). The HPV33:4E5 complex map shows strong density corresponding to five bound 4E5 Fabs around the periphery of the upper rim of the 5-coordinated pentamer of the

capsid; although, much lower and discontinuous densities for the 4E5 Fabs bound to 6-coordinated pentamer were also observed (Fig. 8d, left), which could be mainly caused by steric hindrance among bound Fabs[30]. Analysis of the extracted cryoEM map of one 5-coordinated HPV33 pentamer and one 4E5 Fab revealed that the 4E5 Fab is bound to the outer rim of the HPV33 capsomer at an angle of ~58° to the pentamer surface (Supplementary Fig. 13c). Similar binding occupancies and binding angles (60° for H58-33BC:4E5 and 60° for H58-33BC-52HI:4E5) were observed for 4E5 Fabs bound to L1 monomers of both H58-33BC:4E5 and H58-33BC-52HI:4E5 complexes (Fig. 8d, middle and right, and Supplementary Fig. 13c, middle and right). This suggests that the same recognition mode was adopted by mAb 4E5 to the VLPs of HPV33 and the chimeric HPV58/33 and HPV58/33/52. Further analysis of the mapped epitopes suggests that the 4E5 Fab binding orientation and footprints covering both the BC loop and the EF loop are similar for all three VLPs-Fab complexes (Fig. 8e and Supplementary Fig. 13d). These data provide strong evidence that loop swapping of the BC loop from HPV33 onto the HPV58 basal framework can mimic the molecular surface of HPV33 and give rise to heterotypic neutralizing antibodies in vivo.

## Discussion

Genetic mutation and recombination are routinely used by pathogens to avoid host immune system recognition, particularly in surface-exposed regions which are targeted by the immune machinery; this also happens to be where most of the neutralizing epitopes are located. The immunodominant epitopes of HPV pathogens usually involve multiple surface regions that

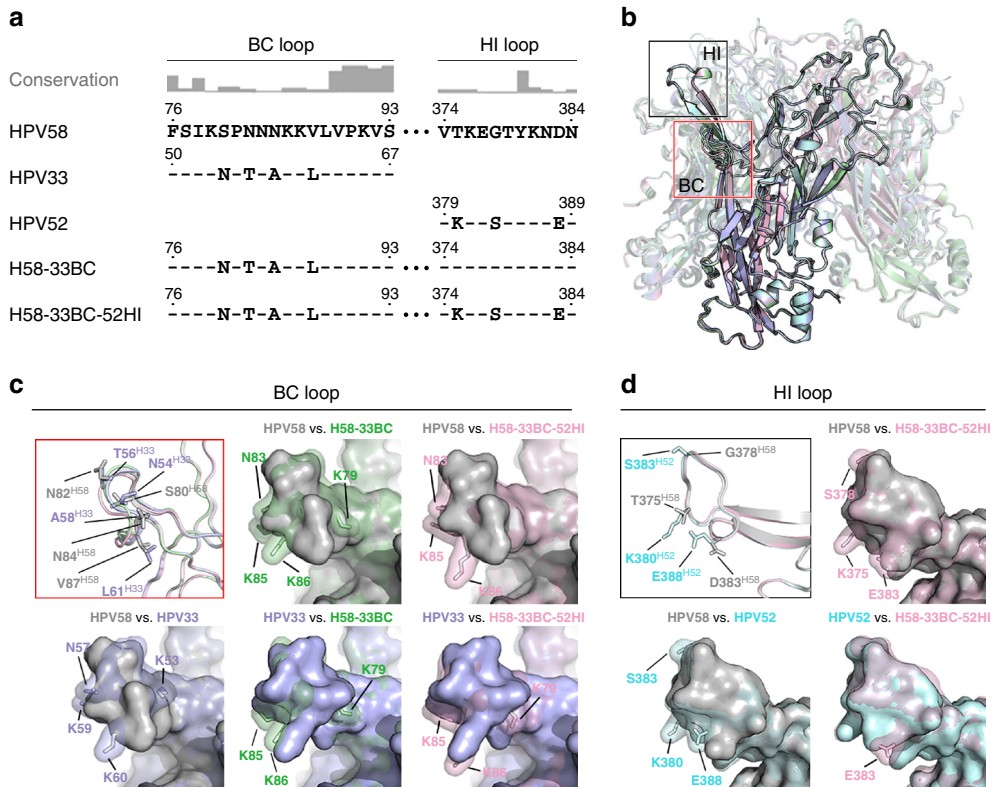

**Fig. 7** Crystal structures of the chimeric pentamers. **a** The modified L1 sequence on the BC and HI loop(s) of the lead candidates of the double- and triple-type chimeric VLPs: H58-33BC and H58-33BC-52HI. The consensus is determined based on an alignment of L1 sequences from 20 different HPV serotypes (also see Supplementary Fig. 4). **b** A structural comparison of pentamers among HPV58 (gray), HPV33 (light blue), HPV52 (pale cyan), and the chimeric proteins H58-33BC (pale green) and H58-33BC-52HI (light pink). Loops BC and HI are boxed in red and black, respectively. **c** Structural comparisons of the BC loops of WT (HPV58 [gray], HPV33 [light blue]) and chimeric L1 proteins (H58-33BC [pale green], H58-33BC-52HI [pale pink]). Upper left: Ribbon display of superimposed main-chain carbon atoms. Side-chains of the different amino acids between HPV58 and HPV33 are labeled and shown as sticks. Upper middle, right and lower: Pairwise structural comparisons with surface display among HPV58, HPV33, H58-33BC and H58-33BC-52HI. **d** Structural comparisons of HI loops of WT (HPV58 [gray], HPV52 [pale cyan]) and the chimeric L1 protein (H58-33BC-52HI [light pink]). Upper left: Ribbon display of superimposed main-chain carbon atoms. Side-chains of the different amino acids between HPV58 and HPV52 are labeled and shown as sticks. Upper right and lower: Pairwise structural comparisons with surface display among HPV58, HPV52 and H58-33BC-52HI. For presentation clarity, the surface of the chimera in the pairwise surface comparisons in **c**, **d** is shown with 50% transparency and embracing the model rendered in stick mode; the surface of the wild-type model is shown with full opaqueness

are discontinuous in the primary sequence, so that type specificity persists even following one or a few amino acids substitutions[22–27,34]. Based on our immunogenicity assays in mice and non-human privates, we have successfully generated triple-type chimeric HPV proteins that can simultaneously present three independent immunodominant epitopes, using seven or fewer amino acid substitutions. We believe that our success is borne from an understanding of the structural and ancestral relationships among multiple HPV types, and, thereafter, the selection of closely related types for epitope grafting.

Our data show that different loops have variable effects on transferring cross-protection between different groups (Figs 4, 5); this could be primarily caused by their inconsistent antigenic determinants for type specificity. Still, additional structural information of the chimeric VLPs needs to be considered to rationalize how to select particular loops for chimeras with high cross-protective efficacy. We also noticed that, among the five surface loops, there is no EF loop-exchange constructs for any of the lead triple-type candidates among the five groups (Group-1, -2, 3, -5 and -6) (Fig. 5), even though an increased heterotypic neutralization could be observed in EF loop-swapped constructs; i.e., the anti-HPV58 antibody titer of H33-58EF was ~2- to 3-log higher than that found following immunization with WT HPV33

VLPs (Fig. 3a). One possible explanation is that the EF loop is not a major component in determining HPV type-specificity. Alternatively, it is possible that the higher flexibility displayed on the EF loop than the other loops makes it more difficult to provide a similar heterotypic functional area through EF loop exchange.

Although the chimeric HPV VLPs had similar physical properties to that of WT VLPs (Supplementary Figs. 8, 9), the chimeric VLPs provide a major increment in HPV protection by inducing efficient neutralizing antibodies against a broad spectrum of HPV, with five chimeric VLPs able to induce immunity against a total of 15 HPV types in mice presented in our study. Even though evolutionary changes tend to be slow in papillomavirus[43,44], recombination events between different types of viruses may lead to vaccine evasion; for example, as mentioned earlier, recombination events between HPV16 and HPV31 can result in a virus carrying HPV16 oncogenes but coding for HPV31 structural proteins or a completely new serotype[21]. Therefore, a cross-type HPV vaccine with a much broader spectrum against similar types could become an alternative means to provide potential protection against the effects of mutation events and/or infection of parental types not included in current vaccines. We propose that, with necessary modifications, an improved HPV vaccine could be engineered to

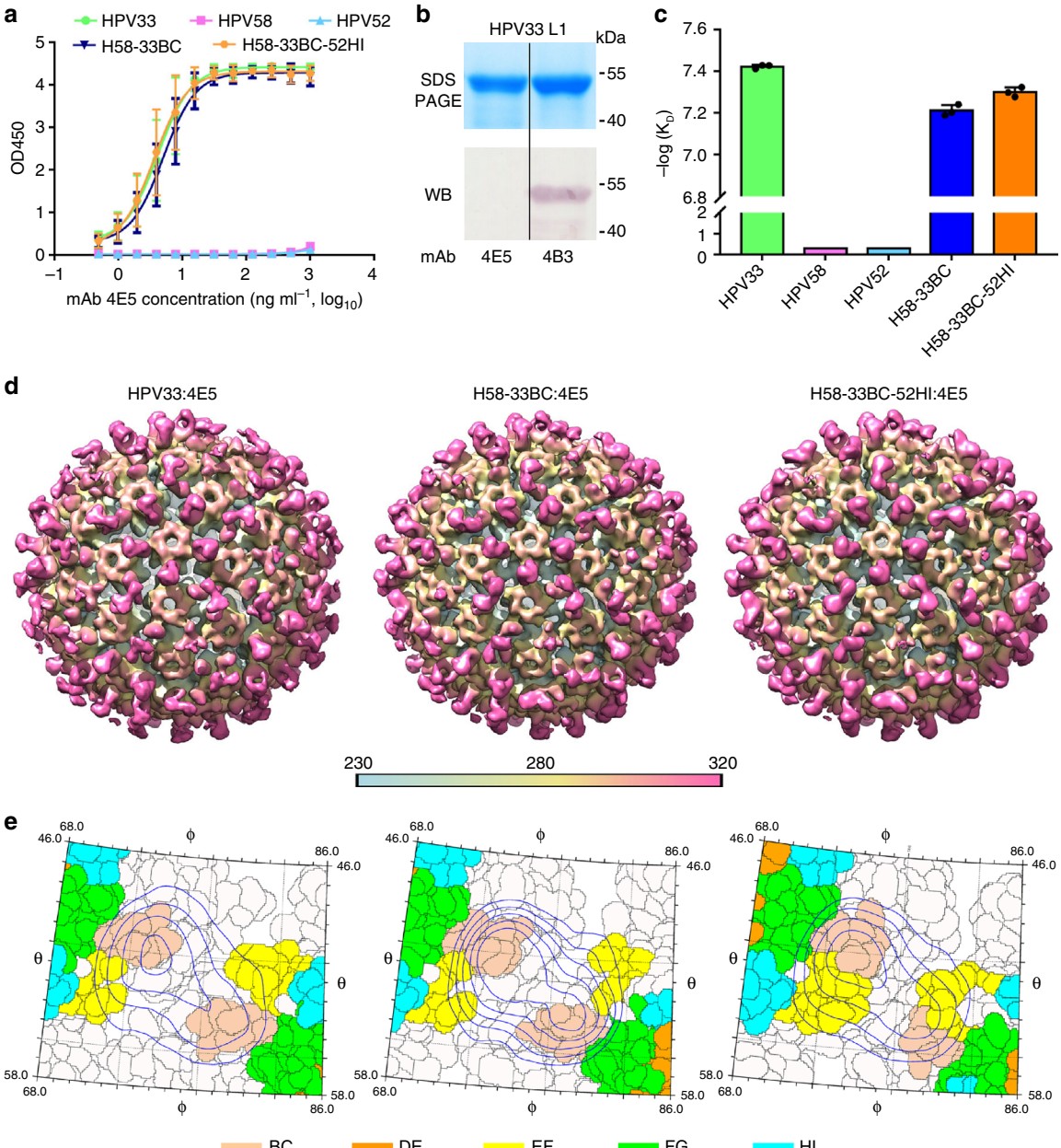

**Fig. 8** Structural characterization of the chimeric pentamers and VLPs-Fab immune complexes. **a** Binding profiles of mAb 4E5 to chimeric mutant and WT VLPs of HPV33, HPV52 and HPV58. Data indicates mAb 4E5 can bind to WT VLPs of HPV33 and chimeric VLPs of H58-33BC and H58-33BC-52HI but fails to bind to WT VLPs of HPV52 and HPV58. The error bars are standard deviation of experiments performed in duplicate. **b** HPV33 L1 protein was subjected to reducing SDS-PAGE and western blot (WB) analysis with mAb 4E5. mAb 4B3, a linear antibody capable of recognizing HPV L1 protein, was employed as a positive control for the WB experiment. **c** Affinity constants of the chimeric mutants (H58-33BC and H58-33BC-52HI) and WT VLPs (HPV33, HPV52 and HPV58) against mAb 4E5 were measured by surface plasma resonance (SPR) in a Biacore 3000. The error bars are standard deviation of experiments performed in triplicate. **d** CryoEM structures of the VLPs-Fab complexes of HPV33:4E5 (left), H58-33BC:4E5 (middle) and H58-33BC-52HI:4E5 (right). Density maps are colored by radius from cyan (230 Å), to yellow (280 Å), to pink (320 Å). Icosahedral 2-, 3- and 5-fold axes are indicated by black symbols. For the purpose of comparability, the cryoEM maps at higher resolution were low-pass filtered to 12.3 Å. **e** Close-up view of the roadmaps of HPV33:4E5 (left), HPV58-33BC:4E5 (middle) and HPV58-33BC-52HI (right) surfaces, projected by an area surrounding a 5-fold vertex defined with a polar angle θ range of 46° to 58° and a polar angle φ range of 68°–86°. Color scheme for surface residues are according to surface loops (BC, amber; DE, orange; EF, yellow; FG, green and HI, Cyan). Blue contour lines were drawn according to the projection of the Fab-4E5 entity with a density greater than 1 and within a radial section of the VLP-4E5 complex ranging from 310 Å to 360 Å. The icosahedral symmetry axes are labeled. The roadmaps were generated by the software RIVEM[68]

elicit protective immunity against 20 types of HPV through seven chimeric VLPs, along with carcinogenic HPV types from group 4, and other low-risk types (e.g., HPV6 and HPV11, which are not associated with cervical cancer, but are linked with precancerous or dysplastic lesions and genital warts) (Fig. 1a). This would hypothetically cover more than 99% of HPV infection that cause cervical cancers[45].

Increasing the immunity of conserved neutralizing epitopes offers a logical strategy for the design of broadly protective immunogens; this would be based on a molecular understanding

of the interactions between isolated broadly neutralizing antibodies and protective antigens[4]. However, no immunogen to date has successfully elicited antibodies with high potent cross-type neutralizing ability, probably due to the limited information available about the conserved regions of natural antigens or the poor immunogenicity of the conserved areas, since such regions appear to be generally inaccessible in the pathogen's native state. Here, we provide a rational approach for structure-based cross-type vaccine design: (1) Structure similarity (where the structure is available) and evolutionary distance analyses were performed to define type groups that could be incorporated into one chimera; (2) Chimeric loop-substituted mutants created between two most closely related types were initially generated to determine candidate(s) that would elicit an ideal double-type immunization response; constructs where swapping occurred between more structurally similar loops were prioritized, as only limited constructs could be generated and assayed for immunogenicity; (3) The remaining four loops of the double-type candidate(s) were then independently exchanged to the corresponding loop of a third genetically close type, and immunogenicity was evaluated to achieve the optimal triple-type chimeric VLPs; again, constructs where swapping had occurred on more structurally similar loops were prioritized. Although HPV sequence diversity is not as high as that of RNA viruses, such as the human immunodeficiency and influenza viruses, or single-stranded DNA viruses, with extensive structural information, the method provided here still could markedly benefit the development of a new generation of vaccines for a wide variety of pathogens.

## Methods

**Protein cloning, expression and purification**. N-terminally truncated HPV16 (GenBank: ANA05496 [https://www.ncbi.nlm.nih.gov/protein/ANA05496]), HPV18 (AAQ92369 [https://www.ncbi.nlm.nih.gov/protein/AAQ92369]), HPV31 (P17388, [https://www.ncbi.nlm.nih.gov/protein/P17388]), HPV33 (AMY16565, [https://www.ncbi.nlm.nih.gov/protein/AMY16565]), HPV35 (P27232, [https://www.ncbi.nlm.nih.gov/protein/P27232]), HPV39 (P24838, [https://www.ncbi.nlm.nih.gov/protein/P24838]), HPV45 (P36741, [https://www.ncbi.nlm.nih.gov/protein/P36741]), HPV52 (AML80965, [https://www.ncbi.nlm.nih.gov/protein/AML80965]), HPV53 (NP_041848, [https://www.ncbi.nlm.nih.gov/protein/NP_041848]), HPV56 (P36743, [https://www.ncbi.nlm.nih.gov/protein/P36743]), HPV58 (AFS33402, [https://www.ncbi.nlm.nih.gov/protein/AFS33402]), HPV59 (CAA54856, [https://www.ncbi.nlm.nih.gov/protein/CAA54856]), HPV66 (ABO76893, [https://www.ncbi.nlm.nih.gov/protein/ABO76893]), HPV68 (AGU90787, [https://www.ncbi.nlm.nih.gov/protein/AGU90787]) and HPV70 (P50793, [https://www.ncbi.nlm.nih.gov/protein/P50793]) L1 genes were cloned into pTO-T7 vector. Mutations were constructed using Gibson assembly[46], a potent reconstitution polymerase chain reaction (PCR) strategy. Briefly, two gene fragments from their templates were cloned by specific primers (Refer to the primer list in Supplementary Data 1). Gibson assembly was then used to construct the targeted clone with these two fragments as templates. All recombinant HPV L1 proteins were produced in ER2566 E. coli strain as described previously[12]. Briefly, the E. coli strain containing L1 expression vectors was induced with 0.4 mM isopropyl-β-D-thiogalactopyranoside for 10 h at 25 °C. The harvested cells were resuspended in buffer (50 mM Tris-HCl, pH 7.2, 10 mM EDTA, 300 mM NaCl) and disrupted by sonication. The supernatant was purified by two chromatography procedures using SP sepharose (GE Healthcare, America) and CHT II resin (Bio-Rad, America). For VLP preparation, the purified L1 proteins were dialyzed with the above-mentioned buffer without reductant to allow VLP self-assembly; To prepare pentamers for crystallization, the Cys175 (as numbered according to the sequence of HPV16 L1) in L1 proteins was replaced with Ser and the mutated proteins were then subjected to limited trypsin digestion (mass ratio of trypsin vs. L1 protein was approximately 1:1000) at room temperature overnight and purified by Superdex 200.

**Crystallization and structural determinations**. Purified HPV L1 pentamers of concentration at ~5 mg mL$^{-1}$ were performed for crystallization trial in sitting-drop vapor diffusion method in 96-well plates at 20 °C. Best crystals were grown by mixing 1 μL sample with 1 μL reservoir solution using the hanging-drop vapor-diffusion method in 24-well plates (Reservoir solutions: for HPV33 pentamer: 0.2 M magnesium formate, 13.5% (w/v) PEG 3350; for HPV52 pentamer: 8% tasimate, 17.5% (w/v) PEG 3350; for HPV58-33BC pentamer: 0.2 M potassium thiocyanate, 23% (w/v) PEG 3350; for HPV58-33BC-52HI pentamer: 0.2 M calcium acetate, 18% (w/v) PEG 3350). Crystals were cryo-protected in their respective reservoir

solutions supplemented with 30% glycerol, and flash-cooled at 100 K.Diffraction data from the HPV33, HPV52, HPV58-33BC, HPV58-33BC-52HI L1 pentamer crystals were collected at the Shanghai Synchrotron Radiation Facility beamline BL17U using a Quantum-315r CCD Area Detector. All datasets were indexed, integrated, scaled and merged using the HKL-2000 program package (http://www.hkl-xray.com). The structures were solved by molecular replacement with PHASER[47]. The crystal structure of HPV58 pentamer (PDB 5Y9E [http://dx.doi.org/10.2210/pdb5Y9E/pdb]) served as the search model. The interest models were built manually in Coot, refined by PHENIX[48] and analyzed by MolProbity[49] (96.3% favored, 0.3% outliers for HPV33; 96.0% favored, 0.2% outliers for HPV52; 97.0% favored, 0.2% outliers for H58-33BC; 95.4% favored, 0.2% outliers for H58-33BC-52HI). Briefly, one round of rigid-body refinement was performed after molecular replacement. Afterwards, the resulting models were manually corrected and refined by COOT[50]. Coordinates and individual B factors of HPV52 and HPV58-33BC pentamers were refined in reciprocal space without non-crystallographic symmetry (NCS) restraints, whereas the coordinates and group B factors of HPV33 and HPV58-33BC-52HI pentamers were refined in reciprocal space with NCS restraints and secondary structure restraints to avoid overfitting. Data collection and structure refinement statistics are given in Table 1.

**Sequence alignment and phylogenic tree construction**. Multiple sequence alignment of the L1 proteins from 20 different HPV types was conducted using Clustal Omega[51] and the phylogenic tree was constructed by the neighbor-joining method using software MEGA 5[52] (http://www.megasoftware.net/). The tree was drawn to scale, with branch lengths measured in the number of substitutions per site. The five additional types of HPV L1 protein sequences downloaded from GenBank (accession numbers: HPV6, AAC80442 [https://www.ncbi.nlm.nih.gov/protein/AAC80442]; HPV11, P04012 [https://www.ncbi.nlm.nih.gov/protein/P04012]; HPV26, NP_041787 [https://www.ncbi.nlm.nih.gov/protein/NP_041787]; HPV51, AJS10540 [https://www.ncbi.nlm.nih.gov/protein/AJS10540]; HPV69, ALJ32828 [https://www.ncbi.nlm.nih.gov/protein/ALJ32828]) along with 15 types of sequences as mentioned above were used for phylogenetic analysis.

**SDS-PAGE and western blotting**. SDS-PAGE was performed using the Laemmli method with minor modifications[53]. Briefly, samples were diluted with Laemmli sample buffer (0.0625 M Tris-HCl, pH 6.8, 2% (w/v) SDS, 10% (w/v) glycerol, 100 mM dithiothreitol, and 0.001% (w/v) bromophenol blue) to a final protein concentration of 1 mg mL$^{-1}$ or 0.2 mg ml$^{-1}$. Samples were heated to 80 °C for 10 min, loaded into the wells of 10% acrylamide gels, electrophoresed, and stained with Coomassie Brilliant Blue.

For western blotting, resolved proteins were electrically transferred to nitrocellulose membranes for 55 min at 35 mA. Membranes were blocked with 5% skim milk, incubated with anti-HPV L1 linear mAb 4B3 (recognizing a broad-spectrum linear epitope, 1:1000 dilution), washed, and then incubated with goat anti-mouse alkaline phosphatase-conjugated antibody (Dako, Denmark). Color was developed over 5 min using NBT/BCIP reagent (Pierce Biotechnology; Rockford, IL).

**Indirect ELISA**. The wells of 96-well microplates were coated with wild type (WT) or chimeric VLPs (300 ng per well) and then blocked overnight at 4 °C. The wells were then incubated with two-fold serial dilutions of the antibody for 45 min at 37 °C. Antibody titers were detected using a horseradish peroxidase (HRP)-conjugated goat anti-mouse IgG antibody (diluted 1:5000 in HS-PBS, Abcam; Cambridge, UK), followed by 50 μL of 3, 3′, 5, 5′-Tetramethylbenzidine Liquid Substrate (Sigma-Aldrich, St Louis, MO) per well for 15 min at 37 °C. The reaction was stopped by the addition of 50 μl of 2 M H$_2$SO$_4$, and the absorbance read at 450 nm (reference, 620 nm) using an automated ELISA reader (TECAN, Männedorf, Switzerland). The median effective concentration (EC$_{50}$, ng mL$^{-1}$) is defined as the antibody concentration for achieving 50% binding with the antigen. Data analysis was performed using GraphPad Prism (GraphPad Software, San Diego, CA).

**Analytical ultracentrifugation (AUC)**. The homogeneity and molecular mass of HPV L1 proteins were determined using sedimentation velocity (SV) experiments. SV was performed at 20 °C on a Beckman XL-A analytical ultracentrifuge, equipped with absorbance optics and an An60-Ti rotor (Beckman Coulter; Fullerton, CA). All the samples were diluted to an OD$_{280nm}$ of 1 in a 1.2-cm light path. A total of 150 scans for samples were recorded and the rotor speed was set to 7,000 rpm for VLPs, and 30,000 r.p.m. for pentamers. The sedimentation coefficient was obtained using the c(s) method with the Sedfit software[54], kindly provided by Dr. P. Schuck at the National Institutes of Health (Bethesda, MD).

**High-performance size-exclusion chromatography (HPSEC)**. Purified WT or chimeric HPV L1 VLPs were chromatographed using a 1120 Compact LC HPLC system (Agilent Technologies; Santa Clara, CA) separately, equipped with a TSK Gel PW5000xl 7.8 × 300 mm column (TOSOH, Japan); columns were pre-equilibrated in phosphate buffer, pH 6.5 with 500 mM NaCl. The column flow rate and protein signal detection for the SEC analysis were 0.5 ml/min and 280 nm, respectively.

**Differential scanning calorimetry (DSC).** DSC was performed on HPV L1 VLPs using a MicroCal VP-DSC instrument (GE Healthcare, MicroCal Products Group; Northampton, MA). Samples (0.5 mg mL$^{-1}$) were measured at a heating rate of 1.5 °C min$^{-1}$ using a scanning temperature that ranged from 10 °C to 90 °C. Software MicroCal Origin 7.0 (Origin-Lab Corp., Northampton, MA) was used to calculate the melting temperatures (Tm) based on the melting curves.

**Monoclonal antibodies (mAbs).** BALB/c mice were immunized subcutaneously three times at 2-week intervals with HPV33 VLPs or HPV58 VLPs (100 μg per mice) absorbed with aluminum adjuvant for prime immunization. Mice also received two additional boost inoculations. Fusion was performed 2 weeks after the final immunization. The resulting hybridomas were screened using indirect HPV33 or HPV58 VLP-based ELISA and pseudovirus-based neutralization assays (PBNA, see below). Positive cells were cloned by limiting dilution at least three times until a single cell clone was attained. MAbs were prepared by injecting hybridoma cells into the peritoneal cavities of pristane-primed BALB/c mice; ascitic fluid was collected after 9–12 days, and the mAbs were purified by Protein A affinity chromatography. The purified anti-HPV33 mAbs (4E5, 7C12, 11A3, 11F4, C2C3, C14A2, C16D7, 9A3 and 7A3) and anti-HPV58 mAbs (4A2, 13A4, 5H2, 10B11, 5G9, A15A9, A10B8, 1D4 and 3E4) were diluted to 1.0 mg ml$^{-1}$ in PBS and stored at −20 °C. mAb concentration was determined with a BCA method.

**BIAcore biosensor analysis.** CM-5 sensor chips were amine-coupled to a goat anti-mouse antibody Fc fragment (GAM-Fc) (BIAcore 3000, GE). One flow cell of a chip was coated with 15,000 RU (resonance units) of the GAM-Fc, whereas the other flow cell was left uncoated and blocked as a control. The affinity measurements of mAb 4E5 binding with HPV33, HPV58, HPV52, H58-33BC and H58-33BC-52HI pentamer, respectively, were initiated by passing HBS (10 mM HEPES, pH 7.4 and 150 mM NaCl) over the sensor surface for 100 s at 30 μl min$^{-1}$, followed by injection of 1 μg ml$^{-1}$ of mAb 4E5 at 30 μl min$^{-1}$ for 3.3 min, and then injections of serially diluted antigens at 30 μL min$^{-1}$ for 3.3 min. Every measurement on the BIAcore 3000 biosensor was performed three times and the individual values were used to produce the mean affinity constant and standard deviation.

**Preparation of HPV pseudoviruses.** The L1/L2 expression vector and pN31-EGFP used in the experiment were kindly provided by Dr. J. T. Schiller[55]. Briefly, plasmids carrying codon-optimized HPV L1 genes were individually co-transfected with an L2 expression plasmid and the marker plasmid into 293FT cells, which were purchased from Thermo Fisher (Catalogue number: R70007) and had been authenticated by Short Tandem Repeat (STR) assay and single nucleotide polymorphism (SNP) genotyping, and tested for the absence of mycoplasma contamination. The cells were harvested 72 h after transfection, lysed with cell lysis buffer containing 0.5% Brij58 (Sigma-Aldrich), 0.2% benzonase (Merck Millipore; Darmstadt, Germany), 0.2% PlasmidSafe ATP-Dependent DNase (Epicenter Biotechnologies, Madison, WI) and DPBS-Mg solution, and incubated at 37 °C for 24 h. Afterward, 5 M NaCl solution was added to the samples to extract the cell lysates. The TCID$_{50}$ (tissue culture infective dose) of the supernatant was then measured to determine the titers of the PsVs, calculated according to the classical Reed–Muench method[56]. Maturation and purification of these samples followed the procedures described before[57–59].

**Pseudovirion-based neutralization assay (PBNA).** 293FT cells were incubated at 37 °C in the wells of a 96-well plate at a density of $1.5 \times 10^4$ cells per well for 6 h. Sera were 2-fold diluted and PsVs were diluted to $2 \times 10^5$ TCID$_{50}$/μL. Equal volumes (60 μl) of the PsV diluent and the serially diluted sera were mixed and incubated at 4 °C for 1 h. The negative control was prepared by mixing 60 μl of the PsV diluent with an equal volume of culture medium. Then, 100 μL of each mixture was added to designated wells and incubated at 37 °C for 72 h. The neutralization titers were calculated as the log$_{10}$ of the highest sera dilution with a percentile of infection inhibition higher than 50%. Every sample was repeated at least three times, and the mean values and SD are reported.

**Ethics statement.** Animal experiments in mice were approved by Xiamen University Laboratory Animal Center (XMULAC). The immunization and serum harvest of cynomolgus macaques were conducted at WuXi AppTec Co., Ltd. (Suzhou, P. R. China) (Contract no. XMU-20160525A). The manipulation protocol was approved by WuXi Institutional Animal Care and Use Committee (IACUC). All procedures were conducted in accordance with animal ethics guidelines and approved protocols to minimize suffering during vaccination, blood collection, and euthanasia.

**Animals, immunizations and serological analysis.** To initially assess the immunogenicity of all chimeric VLPs, special pathogen-free (SPF) BALB/c mice (n = 4 or 5) were immunized intraperitoneally three times at an interval of 2 weeks (weeks 0, 2 and 4) with chimeric or WT VLPs diluted in aluminum adjuvant (5 μg per dose). Serum samples were collected at week 8, and the neutralizing titers were analyzed by PBNA, as described above.

To verify cross protection of H33-59HI and H58-59BC chimeric VLPs, WT VLPs and chimeric H33-58HI and H58-33BC VLPs (100 μg per dose), absorbed with Freund's adjuvant, were used to immunize SPF BALB/c mice (n = 3) subcutaneously three times at an interval of 2 weeks (weeks 0, 2 and 4). Serum samples were taken at week 8 after the first immunization to determine the neutralizing titers by PBNA.

For dose-dependent response immunogenicity evaluations, SPF BALB/c mice were immunized intraperitoneally three times with three different doses (0.1, 1 and 10 μg, n = 5 or 6 per dose) at an interval of 2 weeks (weeks 0, 2 and 4) with chimeric or WT VLPs diluted in aluminum adjuvant. Serum samples were collected at week 8 for PBNA.

For ED$_{50}$ calculations, 40 SPF BALB/c mice (n = 8 per group) were vaccinated by intraperitoneal injection with a single dose (0.3, 0.1, 0.033, 0.011, or 0.004 μg) of each chimeric or WT VLP (serially diluted in aluminum adjuvant). The dose of each VLP in the mixed WT VLP group was equal to that of the same VLP in the single-type groups. Serum samples were taken 4 weeks after immunization, and the seroconversion at each dose was calculated. The ED$_{50}$ was reported in units of micrograms of antigen per mouse, resulting in 50% seroconversion. The ED$_{50}$ results were analyzed based on the dose-response curve using a Reed–Muench model[60].

To assess the immunogenicity of H58-33BC-52HI VLPs in non-human primates, 20 cynomolgus macaques (Macaca Fascicularis)—with no detectable neutralizing antibodies against infection of HPV33, -52 or -58—were divided into five groups (n = 4). The first group was inoculated with H58-33BC-52HI VLP (5 μg per dose); the other four groups were inoculated with HPV33 (5 μg per dose), HPV52 (5 μg per dose), HPV58 (5 μg per dose) VLPs, and the mixed VLPs of HPV33/52/58 (5 μg of HPV33, -52 and -58 per dose), respectively. All five groups were immunized twice (months 0 and 2) by Intramuscular injection. Pre-immune serum samples of each monkey were collected individually, and serum samples were collected at 2-week intervals after immunization. The titers of HPV neutralizing antibodies in these sera were detected by PBNA. All data for the neutralization titers of per group analyzed by PBNA are plotted as the mean with SD.

The sample size of animal studies were estimated according to our previous studies[61]. All the mice used in this study were female and about 4–6 weeks old and the age of mice were much the same in the same set experiments. The male and female cynomolgus macaques were about 3 years old. All the animals were divided randomly into a certain number groups. No blinding was done.

**Cryo-EM and three-dimensional (3D) reconstruction.** Purified mAb 4E5 Fab fragments were incubated with HPV58, HPV58-33BC, or HPV58-33BC-52HI VLPs at 37 °C for 2 h (molar ratios of 1:1.2 for all immune complexes). Small aliquots (3 μL) of mixtures were deposited onto glow discharged holey carbon Quantifoil Cu grids (R2/2, 200 mesh, Quantifoil Micro Tools), blotted with filter paper for 6 s, and then vitrified by plunging into liquid ethane inside an FEI Mark IV Vitrobot at 100% humidity. CryoEM micrographs were collected at 300 kV with a FEI Tecnai F30 transmission electron microscope, equipped with Falcon II direct electron detector at a defocus level from 1.5 to 3.5 μm evaluated using Gctf[62]. The nominal magnification was set to 93,000 (pixel size = 1.128Å) with an electron dose of 25 e$^-$/Å$^2$ at 1 s. Particles were manually picked using the e2boxer.py program in EMAN2.1[63]. After several rounds of reference-free 2D classifications using Relion2.0[64], good particles were sorted for further 3D reconstruction and refinement with AUTO3DEM[65]. Final map resolutions were determined by the gold standard FSC curve between two-half maps with a cutoff of 0.3[66]. HPV58 pentamer crystal structure were manually fitted using the program Chimera, and further optimized into the corresponding maps using the "fit in map" command in Chimera[67].

**Stereographic roadmap projections of HPV VLP-Fab complexes.** The crystal structures of HPV33, HPV58-33BC, HPV58-33BC-52HI were fitted into the corresponding cryoEM density map using the tool "fit in map" in Chimera, and were then plotted onto stereographic spheres using RIVEM[68] (Radial Interpretation of Viral Electron Density Maps), where the polar angles θ and φ represent latitude and longitude, respectively. The Fab density was projected as contour lines onto the sphere, which is presented as a stereographic diagram.

## Data availability:

The data supporting the findings of this study are available from the corresponding authors upon reasonable request. Atomic coordinates and structure factors for the HPV33, HPV52, H58-33BC and H58-33BC-52HI L1 pentamers have been deposited in the Protein Data Bank (accession code 6IGE, 6IGF, 6IGD, and 6IGC). The cryoEM density maps for HPV33:4E5, H58-33BC:4E5 and H58-33BC-52HI:4E5 have been deposited in the Electron Microscopy Data Bank (accession code EMD-9665, EMD-9666 and EMD-9667). A reporting summary for this Article is available as a Supplementary Information file.

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

## Acknowledgements

We acknowledge use of beamline 17U of the Shanghai Synchrotron Radiation Facility (SRFF) for X-ray diffraction data collection. This work was supported by grants from the National natural Science Foundation of China (No. U1705283, 31670935 and 81701637), the National Science and Technology Major Projects for Major New Drugs Innovation and Development (No. 2018ZX09738008). The funders had no role in the study design, data collection and analysis, decision to publish, or preparation of the manuscript.

## Author contributions

S.L., Y.G. and N.X. designed the study. Z.L., S.S., M.H., D.W., J.S., X.L., Y.L., X.C., S.W., Y.Y., Z.W., J.L., H.Q. and Q.Z. performed the experiments. Z.L., S.S., M.H., D.W., H.Y., X.Y., Y.G., S.L. and N.X. analyzed the data. Z.L. and S.L. wrote the manuscript. Z.L., S.S., M.H., D.W., H.Y., X.Y., Q.Z., J.Z.,Y.G., S.L. and N.X. participated in discussion and interpretation of the results. All authors contributed to experimental design.

## Additional information

**Competing interests:** The authors declare no competing interests.

