## [Peer Review File · Nature Communications]

Reviewers' Comments:

Reviewer #1:

Remarks to the Author:

The manuscript presents an impressively detailed and comprehensive evaluation of chimeric L1 VLP vaccines for oncongenic HPV types. The overall findings can be considered unexpected as previous attempts at swapping exposed loops on the pentamer surface to increase the induction of cross-type neutralizing antibodies had not been successful. The success in this case can be attributed to 1) the use (and in some cases the generation) of high resolution structural data to construct the chimeras, 2) the decision to swap domains among closely related types, and 3) the generation and evaluation of an exceptionally large number of chimeras, since the structural data could not entirely predict which chimeras would be best able to generate cross-type neutralizing antibodies. The generation of double recombinant VLPs that generate titers of neutralizing antibodies against the three targeted types that approach the titers generated by vaccination with a combination of the three wild type VLPs is a remarkable achievement. The obvious advantage of the chimeric VLP approach is that it provides the potential to protect against more types with fewer particle types, thereby decreasing manufacturing complexity and cost, and reducing the amount of protein and adjuvant that needs to be administered. There is a reasonable chance that these chimeric VLPs will be moved forward into a clinical trial, since the E. coli-based production used in this study was used to produce Xiamen Inovax's HPV16/18 vaccine which is currently in a phase 3 trial.

While the study represents a technical tour de force, there are two potential limitations to the approach from the prospective of a widely employed commercial vaccine. First, the current VLP vaccines induce remarkably consistent antibody responses in humans, despite the diversity in human immunoglobulin gene loci. In part, this consistency can be attributed to the fact (strongly supported by the data herein) that there are multiple loops on the VLP surface that can generate neutralizing antibodies, and most individuals generate antibodies to multiple loops. Limiting the potential neutralizing epitopes to a single loop per type may increase the variability in responses, since some individuals may not have a B cell germ line configuration that can generate high avidity neutralizing antibodies to a single specific loop after a limited number of somatic mutations. Second, restricting the neutralizing epitopes to a single loop per type increases the chances that a single or small number of amino acid changes in L1 will generate an immune escape variant for that type.

Specific Comments:

1. Line 25: the stated number oncogenic types varies in the text from 15-18. Please be consistent.
2. Line 34: I don't like the use of the "ultimate" descriptor, even if it is in quotation marks. "Improved" would be preferable.
3. 1st paragraph Introduction: Emphasizing the obstacles imposed by the evolution of antibody escape variants as a primary justification for this vaccination strategy seems a bit off target since chimeric VLP vaccines would be more susceptible to emergence of escape variants than current commercial vaccines for the reason stated above. I would emphasize the potential for increased type coverage with decreased manufacturing complexity, cost, and protein and adjuvant levels as the primary rationale for this strategy.
4. Line 61: protection with the VLP vaccines is not "strictly type-specific", as acknowledged subsequently. It's better to use the term "type-restricted".
5. Line 64: "antibodies are less durable" is unclear. I don't know of data indicating that the decay curves of cross-neutralizing antibodies over time are any steeper than those of type-specific neutralizing antibodies. However, the peak titers are much lower and so potentially may drop below protective levels sooner.
6. Line 85: change to "type-specificity is largely dependent". There are some type specific neutralizing monoclonal antibodies that recognize epitopes in the canyons between pentamer surfaces.

7. Line 89: change "wide" to "wider". Gardasil 9 could easily be consider a wide-spectrum vaccine.
8. Line 111: This is an unintentionally misleading statement. The pentamers make up the entire shell of capsid, not just the surface.
9. Line 162: change to "genetically closely related HPV types" and to "from more distantly related types".
10. Line 204: Please justify the use of IP injection rather than IM, as is used with the human vaccines?
11. Line 224: the dose response curve is very shallow. Is there a significant difference in the titers induced by the three different doses?
12. Line 240: the decision to use an exceptionally high dose and Freund's adjuvant could be explained here. The lack of cross-neutralization with these chimeras is more dramatic by using this potentially highly pro-immunogenic protocol, but some readers may miss the point.
13. Line 244: change "cannot" to "was not". This experiment certainly supports, but does not categorically prove, that chimeric VLPs involving distantly related types don't work.
14. Line 286: Why weren't the BC and HI loops substituted here, as it in the successful experiments detailed abobe? Is there a rationale for selecting these particular loops for these chimeras?
15. Line 290: the neutralizing titers were similar, but it seems very likely that breath of the antibody responses ("repertoire") was narrowed because the epitopes of only a single of the surface loops were presented.
16. In Figure 4B, it would be helpful to add HPV33, HPV52 and HPV 58 above the graphs, or at least indicate which graph corresponds to which virus in the legend.

Reviewer #2:

Remarks to the Author:

The article by Li et al. "Rational Design of a Triple-Type, Non-Env 1 eloped Virus Vaccine by Compromising the Type Specificity of Human Papillomavirus" describes how they performed loop swapping for all the five surface loops of L1 pentamers between a group of three closely related HPV types to enhance the level of cross-type protection to a significant level, even though the neutralization titer is generally lower than the WT HPV VLPs. The data are well presented, and will offer insight to the relative importance of different surface loops in determining the type-specific neutralising epitopes for HPV vaccine. This work will be of interest to those in viral immunology area, and is well suited for a more specialized journal.

I have the following major comments:

Comment 1)

Fig.1 shows the conformation differences of the five surface loops between a few pairs of HPV types. These are backbone conformations, and for the most part, if the loop length is the same (such as 33/58 pair), the loop conformations are identical, with the slight differences being within the variation of the same capsomere of the same structure. However, even in this situation, the surface contour will show difference because of the different side chains between the two types. For example, the -TKEG- of 58 versus -TSDS- of 33 on HI-loop, the surface features of KEG- should differ from those of SDS-. When the loop has different length, the loop conformation and surface contour is expected to be different, as shown by several L1 structured published from multiple groups. Such expected structural comparison results should be discussed. This consideration also can explain most of the efficacy effect of different loop swapping results.

It will also be better if the pairwise loop sequence comparison are shown right below each Figure 1C panels.

Another concern with such comparison is using the loop detached from the entire pentamer. Does any of these loops show larger or smaller difference when superimposed based on an entire pentamer between the two types?

Comment 2)

The aforementioned point is also reflected in Fig. 5, the determination of the chimeric pentamers: H58-33BC and H58-33BC-52HI to compare with the replaced loops, and concluded that BC loops of both H58-33BC and H58-33BC-52HI have the tips orientated $\sim 7 \text{ \AA}$ and $\sim 5 \text{ \AA}$, respectively, away from the corresponding loop of the HPV58 pentamer. To show the different BC loop confirmations of the chimeric pentamers, the native pentamers of H58, 33, and 52 should be compared also in the same way.

Comment 3)

The H33 specific neutralizing mAb 4E5 was used to examine binding to the H58-33BC and H52-33BC-52HI chimera using EM. Because it binds to BC loop, it appears it was the right choice. It is unclear how the authors chose 4E5 for this study, do they already know it target H33-BC loop ahead of time? Did they test other mAbs that are specific for H33?

Comment 4)

The loop swapping has been tried before to show the transfer of respective epitopes for a particular neutralising antibody, although systematically extensively. This study tried swapping of all five loops between different types. Among the 5 surface loops, swapping EF loop has the least effect on transferring the cross-protection. Does that mean EF is the outermost positioned and most flexible loop to be a good neutralizing epitope? It would be good to add some discussion regarding this, and the differences of other loop transferring effect for different type.

Comment 5)

It's hard to guess the paper's content from the current title. Title needs to be changed to reflect more on the actual work.

Minor points:

Page 17, line 356 <5A-5C and S12> Where is S12?

Page 16, line 332: <at levels comparable with that achieved by mixing three WT HPV VLPs at three times the dosage (versus one dose with the chimeric VLPs; Figure 3F)> Where is Figure 3F?

Reviewer #3:

Remarks to the Author:

Chimeric VLP were designed in this work to be used to develop a better HPV vaccine that will prevent infection from a wider group of HPV serotypes and can be administered in a normally sized dose made with standard manufacturing practices. The proposed approach is based on a hypothesis that cross-type protection against HPV infection could be achieved by creating a chimeric HPV VLP. There is an impressive amount of work that has gone into the manuscript. The most compelling result is that all of the chimeric HPV33/58 VLPs showed improved heterotypic antibody responses, and this occurred without changing the elicitation of neutralization antibodies against their own backbone types. This

finding was reinforced by the low ED50 values. However it was disappointing that cross-protective immunities cannot be elicited by proteins engineered using genetically distant types. More importantly though, unfortunately, there is a major problem with the basic premise and some smaller concerns.

Although it seems logical that there is a correlation between structural variation and evolutionary distance among HPV types, the results presented here do not support this basic premise. Using the calculated RMSD as a basis to support this hypothesis is flawed because the structural differences of L1 loops are insignificant. For example the mean RMSDs (for the variable region of paired genotypes) 0.66 Å (HPV33/58), 0.74 Å (HPV58/52), 0.94 Å (HPV11/16) and 2.02 Å (HPV58/59) do not support an increasing difference in structure. In fact quite the opposite is established from these RMSD values; this range actually supports a conclusion that the structural differences are negligible. To provide context, a hydrogen bond length is 2.2-3.2Å and the resolutions of the maps are 2.9 and 2.75Å.

This fallacy is reported again in Sup fig 6 where RMSD differences range from 0.1 - 0.6Å, a range that establishes no significant difference between structures.

Also Line 341-345. There are no remarkable HI loop differences in the structure of the main-chain carbonfrom the manuscript : "there were distinct differences in the surface contours of the (Figure 6C, lower right and Supplementary Figure 11B)." Figure 6C is the wrong figure to show this. SFig11B shows no significant differences in contour.

To be very clear, the structural differences do not support a conclusion that genetically closer types share more structural similarity than distant ones. Thus there is no "strong evidence" based on structural differences to explain the cross protection achieved by HPV vaccines against infection with a limited number of phylogenetically related HPV types. (line 158-160)

Additionally, structural conservation with sequence divergence is a common tenet of structural virology. The concept that genetically closer types share more structural similarity than distant ones, particularly on the outer surfaces is not supported.

Other concerns:

It is unclear what the antibody binding experiments add to our knowledge of HPV antigenic capsid structure. A single point mutation can abrogate binding of a neutralizing antibody, so it is no surprise that a chimeric loop also affects MAb binding.

H33-58HI and H58-33BC capsids are quite heterogeneous. Are they as stable as the others and does this have any bearing on their immunogenicity?

The authors state that residue substitutions on the HPV58 BC and HI loops change the orientation and conformation of the pentamer - this is overstated, somewhat incredible, and not shown in the figure cited. Any changes conferred by point mutations are typically small, likely limited by the side chain movement, but could affect local patch surface charge depending on the substitution. Therefore the conclusion is also overstated as no significant structural alteration has been demonstrated.

For the cryo-EM maps, a central section of each map should be shown to verify the strength of the Fab density relative to the capsid density.

At what sigma level are the surface contoured maps displayed? The capsomers are disconnected. Also, none of the maps appear to be 10-12Å resolution. According to the Methods, Relion was used to classify. These classes should be shown in Supplemental to address the issues of heterogeneity.

It is unclear whether gold standard procedure was followed throughout. The Auto3DEM version cited does not automatically split the data sets prior to reconstruction, which would mean that a FSC with cutoff of 0.5 should be used and not 0.3. Furthermore, auto3DEM can sometimes overestimate resolution, which could also be addressed by including central sections of the cryo-EM maps.

Resolutions for the complex maps are too poor to conclude that the Fabs have identical interactions.

Fig S13, Why does an FSC at 0.3 ensure resolution reliability of a reconstruction with only 100 particles? With the heterogeneity visible in the micrographs it is incredible that 10-12Å was attained with only 100 particles.

Icosahedral symmetry axes need to be added to the FAb-casomer panels (C) to provide context. Otherwise the angle seems arbitrary.

In both Fig 6 and Fig S13 the roadmap needs to be described to explain to a reader how a 3-D map is rendered in 2-D, how were the contoured footprints were projected (and their significance), what section of the virus surface is displayed, and other information. Additionally, the Roadmap program should be properly cited in the legends.

Roadmap use should be described (and cited) in the Methods. What conclusions are drawn from the projected contours of the Fab density? How was this projection done? Why are the surface residues not labeled?

Discussion line 420 needs modification : "feasible approach, based on the knowledge of the intrinsic connection between structural diversity and evolution"

Very minor issues:

Influenza is not an appropriate ssDNA virus to reference in regard to lack of mutational drift.

Response to Reviewer Comments on the manuscript

Manuscript ID: NCOMMS-17-34032-T

We would like to thank all the reviewers for their constructive comments which greatly improved the manuscript. We have addressed all the comments of the reviewers with appropriate additional experiments and analysis.

=====

Reviewer #1

General opinion and comments:

Reviewer: The manuscript presents an impressively detailed and comprehensive evaluation of chimeric L1 VLP vaccines for oncogenic HPV types. The overall findings can be considered unexpected as previous attempts at swapping exposed loops on the pentamer surface to increase the induction of cross-type neutralizing antibodies had not been successful. The success in this case can be attributed to 1) the use (and in some cases the generation) of high resolution structural data to construct the chimeras, 2) the decision to swap domains among closely related types, and 3) the generation and evaluation of an exceptionally large number of chimeras, since the structural data could not entirely predict which chimeras would be best able to generate cross-type neutralizing antibodies. The generation of double recombinant VLPs that generate titers of neutralizing antibodies against the three targeted types that approach the titers generated by vaccination with a combination of the three wild type VLPs is a remarkable achievement. The obvious advantage of the chimeric VLP approach is that it provides the potential to protect against more types with fewer particle types, thereby decreasing manufacturing complexity and cost, and reducing the amount of protein and adjuvant that needs to be administered. There is a reasonable chance that these chimeric VLPs will be moved forward into a clinical trial, since the *E. coli*-based production used in this study was used to produce Xiamen Inovax's HPV16/18 vaccine which is currently in a phase 3 trial.

Response: Thank you to the Reviewer for their encouraging comments. We are excited by your comments and prospective of our triple-type HPV vaccine candidate. Our vaccine design strategy was inspired by our previous work (Li et al, *mBio* 2017), where we showed that the binding of type-specific neutralizing antibodies could be redirected to genetically close HPV types by swapping only a few key epitope residues. By employing this strategy across several triplet combinations of HPV types, we will have 7 chimeras that extends the coverage to at least 20 HPV types, which encompasses all of the proposed oncogenic HPV types (Fig. 1b). From our results, we posit that the production process and assembly procedure would be almost the same for all chimeras as that of their parental-type HPV VLPs. Future work will be based on our well-established *E. coli*-based vaccine R&D platform that was previously used in the development of our HPV16/18 bivalent vaccine (finishing phase 3 trial), HPV6/11 bivalent vaccine (in phase 2 trial) and HPV 9-valent vaccine (approved for clinical trial in China). Because of these advantages, the novel triple-type HPV vaccine now is in the preclinical stage and we expect to submit an Investigational New Drug (IND) application next year.

While the study represents a technical tour de force, there are two potential limitations to the approach from the prospective of a widely employed commercial vaccine. First, the current VLP vaccines induce remarkably consistent antibody responses in humans, despite the diversity in human immunoglobulin gene loci. In part, this consistency can attributed to the fact (strongly supported the data herein) that there are multiple loops on the VLP surface that can generate neutralizing antibodies, and most individual generate antibodies to

multiple loops. Limiting the potential neutralizing epitopes to a single loop per type may increase the variability in responses, since some individuals may not have a Ig germ line configuration that can generate high avidity neutralizing antibodies to a single specific loop after a limited number of somatic mutations. Second, restricting the neutralizing epitopes to a single loop per type increases the chances that a single or small number of amino acid changes in L1 will generate an immune escape variant for that type.

Response: Thank you for bringing to our attention this frequent concern with epitope vaccines. As mentioned by the Reviewer, the major neutralization determinants of the HPV L1 capsid are conformational and tend to comprise parts of the five surface loops of HPV L1 instead of a single loop therein or a so-called linear peptide; for example, the HPV16-V5 epitope involves BC, DE, FG and HI loops (*Lee, Brendle et al. J. Virol. 2014*); HPV16-1A involves DE, FG, and HI loops (*Guan, Bywaters et al. Virology.2015*); HPV16-H263.A2 involves DE, EF, FGs and HI loops (*Guan, Bywaters et al. Virology. 2015*), HPV-14] involves all five surface loops (*Guan, Bywaters et al. Virology. 2015*); HPV58-A12A3 involves DE and FG loops (our previous study, refer to *Li, Wang et al. mBio 2017*); HPV33-4E5 involves BC and EF loops (this study, Fig. 6e). In this regard, the grafted epitopes represented by single loop swapping between two close types may involve not only the substituted loop per se but also cover other neighboring loops, and therefore, the overall structural context of the parental-type capsid protein L1; this hypothesis is supported by HPV33-4E5 binding to the HPV58/33 chimera via not only the BC loop but also the EF loop in our study. However, this would not be the case for loop swapping between two *distant* types; for example, BC loop and/or HI loop swapping of HPV58 cannot reconstruct the neutralization epitopes of HPV59 (Figure S7), as occurred in another study, where a low level of cross-protection was generated for HPV11/16 (*Christensen, Cladel et al. Virology.2001*). This lack of overlap may be because the epitope of the distant type cannot be resurfaced with a single loop implanted within a different structural context (different neighboring loops and diverse structural context). In summary, our results suggest that the HPV antisera spectrum related to type-restricted neutralization can be ascribed two aspects: 1) the overall structural context of the major neutralization regions: the more genetically close in genotype, the more similar context; and 2) type-specific aa sequences in the surface loops: neutralization will stem from substitutions that can convert and/or compromise the type-restricted immunogenicity of local neutralization regions between two HPV types. Although we have not measured the ratio of the type-specific antibodies and the potential cross-type antibodies in the antisera induced by the HPV58/33/52 triple-type chimera, neither of these neutralizing antibodies can be induced by a single loop, and thus the first potential limitation raised by the Reviewer may not apply to our triple-type VLP vaccine.

As to the second potential limitation, most of the HPV neutralization epitopes comprise multiple loops and may have tolerance against the mutation of a single or small number of epitope residue(s). Moreover, the sites associated with type-restriction have less conservation among HPV isolates (Figure S10). Additionally, every residue associated with type-restricted neutralization contributes to cross-neutralization when substituted to the other type. For instance, there are four different residues between the BC loops and HI loops of HPV58/33. The single mutation chimeras of HPV33 VLPs (mutated to the HPV58 HI loop, H33-S350K, H33-D351E, H33-S352G, H33-E357D,) and HPV58 VLPs (mutated to the HPV33 BC loop, H58-S80N, H58-N82T, H58-N84A, H58-V87L) can elicit cross-neutralization antibodies (Figure R1 below), which suggests that there is a very rare chance that multiple mutations simultaneously happening on all of the type-restricted residues could lead to immune escape, as suggested.

Figure R1: Immunogenicity of single-point substituted HPV33/58 chimeric VLPs.

Groups of mice (n = 4) were immunized with 5 µg/dose of the selected chimeric or WT HPV33/58 VLPs at weeks 0, 2 and 4. Immune sera were collected at week 8 and analyzed by neutralization assays.

Specific comments (numbers in reviewer section indicate lines in the original manuscript):

Comment 1: Line 25: the stated number oncogenic types varies in the text from 15-18. Please be consistent.

Response: The number of oncogenic types is now consistently referred to as “18” throughout this manuscript. Thank you for pointing out this error (Page 3, Line 53)

Comment 2: Line 34: I don't like the use of the "ultimate" descriptor, even if it is in quotation marks. "Improved" would be preferable.

Response: As suggested, “ultimate” has been changed to “improved”. (Page 2, line 31)

Comment 3: 1st paragraph Introduction: Emphasizing the obstacles imposed by the evolution of antibody escape variants as a primary justification for this vaccination strategy seems a bit off target since chimeric VLP vaccines would be more susceptible to emergence of escape variants that current commercial vaccines for the reason stated above. I would emphasize the potential for increased type coverage with decreased manufacturing complexity, cost, and protein and adjuvant levels as the primary rationale for this this strategy.

Response: As suggested, we have modified the Introduction to underscore this consideration: “...However, multiple antigenic types or subtypes derived from the evolution of pathogenic microbes through sequence variation presents a serious obstacle in vaccine development. One way to tackle this variation is to include more antigenic variants into a single vaccine, as exemplified with the *Streptococcus pneumoniae* vaccine and *Human papillomavirus* (HPV) prophylactic vaccine. Yet, because pathogens, such as the influenza viruses and human immunodeficiency virus (HIV), have very high levels of antigenic variability, this approach is fraught with difficulties, as an increase in type coverage will dramatically enhance protein amount and adjuvant level per dose, as well as increase the manufacturing complexity and associated production costs. Studies that focus on”. (Pages 2-3, lines 36-44)

Comment 4: Line 61: protection with the VLP vaccines is not "strictly type-specific", as acknowledged subsequently. It's better to use the term "type-restricted".

Response: “Strictly type-specific” is now changed to “type-restricted”. (Page 4, line 58)

Comment 5: Line 64: "antibodies are less durable" is unclear. I don't know of data indicating that the decay curve of cross-neutralizing antibodies over time are any steeper than those of type-specific neutralizing antibodies. However, the peak titers are much lower and so potentially may drop below protective levels sooner.

Response: As suggested, according to the reference, this sentence now reads “Pre-clinical and clinical data show that vaccinated patients exhibit low titers of neutralizing antibodies against genetically related, non-vaccine HPV types in highly reactive immune sera, and these antibodies may drop below protective levels sooner than type-specific ones.” (Page 4, lines 59-61)

Comment 6: Line 85: change to "type-specificity is largely dependent". There are some type specific neutralizing monoclonal antibodies that recognize epitopes in the canyons between pentamer surfaces.

Response: Thank you for this suggestion. The change has been made. (Page 5, line 80)

Comment 7: Line 89: change "wide" to "wider". Gardasil 9 could easily be considered a wide-spectrum vaccine.

Response: The word “wide” is now changed to “wider”. (Page 6, line 98)

Comment 8: Line 111: This is an unintentionally misleading statement. The pentamers make up the entire shell of capsid, not just the surface.

Response: We apologize for this confusion. We have changed the sentence as follows: “The type-specific neutralization epitopes of HPV, which are located on the surface of the HPV capsid, are mostly mediated by the pentamers”. (Page 6, lines 101-102)

Comment 9: Line 162: change to "genetically closely related HPV types" and to "from more distantly related types".

Response: As suggested, the revised sentence now reads: “We thus proposed that highly potent cross-type protection against HPV infection could be achieved by creating a chimeric HPV VLP that contains antigenic determinants from genetically related HPV types rather than from genetically unrelated types.” (Pages 8-9, lines 138-140)

Comment 10: Line 204: Please justify the use of IP injection rather than IM, as is used with the human vaccines?

Response: Yes, we used IP injections with 1mL volume samples in mice to minimize manipulation error; however, we used IM injections in bigger animals, such as rabbits, rhesus monkeys and so on, as the practical immunization used in humans. Please refer to our previous vaccine characterization studies (*Gu, Wei et al. Vaccine. 2017; Pan, Li et al. Vaccine. 2017*).

Comment 11: Line 224: the dose response curve is very shallow. Is there a significant difference in the titers induced by the three different doses?

Response: Due to the excellent immunogenicity of the VLPs, we only can observe significant differences in the neutralization titers in the lowest dose group in this study. Please refer to the statistical analysis below that illustrates the differences among the three doses (Figure R2, below).

Figure R2: Dose-dependent neutralization titers.

The regrouped data drawn in this figure is same as that presented in Figure 2D. Significant differences are compared with the dose-dependent neutralization titer. *P < 0.05; **P < 0.01; ***P < 0.001.

Comment 12: Line 240: the decision to use an exceptionally high dose and Freund's adjuvant could be explained here. The lack of cross-neutralization with these chimeras is more dramatic by using this potentially highly pro-immunogenic protocol, but some readers may miss the point.

Response: Thank you for bringing this consideration to our attention. As suggested, an additional explanation has been added, and the revised manuscript now reads: "To evaluate their potential cross-type immunogenicity, mice were immunized three times with a higher dose (100 µg) of proteins absorbed with a stronger Freund's adjuvant". (Page13, lines 218-219)

Comment 13: Line 244: change "cannot" to "was not". This experiment certainly supports, but does not categorically prove, that chimeric VLPs involving distantly related types don't work.

Response: The word "cannot" has been changed to "was not". (Page 13, line 222)

Comment 14: Line 286: Why weren't the BC and HI loops substituted here, as it in the successful experiments detailed above? Is there a rationale for selecting these particular loops for these chimeras?

Response: The rationale for selecting loops for these chimeras is three-fold: (1) Structure similarity (where the structure is available) and evolutionary distance analyses were performed to define type groups that could be incorporated into one chimera; (2) Chimeric loop-substituted mutants created between two most closely related types were initially generated to determine candidate(s) that would elicit an ideal double-type immunization response; constructs where swapping occurred between more structurally similar loops were prioritized, as only limited constructs could be generated and assayed for immunogenicity; (3) The remaining four loops of the double-type candidate(s) were then independently exchanged to the corresponding loop of a third genetically close type, and immunogenicity was evaluated to achieve the optimal triple-type chimeric VLPs; again, constructs where swapping had occurred on more structurally similar loops were prioritized. In our study, although BC- and HI loop-substituted chimeras exhibited cross-protection for Groups-2, -3, -5 and -6 in the primary screening, we only present the best candidates in the manuscript: these candidates were finally selected by their highest neutralization titers for all three types in the immunogenicity assays. We hope this clarifies our rationale for the reviewer.

Comment 15: Line 290: the neutralizing titers were similar, but it seems very likely that breath of the antibody responses ("repertoire") was narrowed because the epitopes of only a single of the surface loops were presented.

Response: As suggested, we revised the statement and now it reads: "... without significantly affecting the original immunization response." (Page 16, lines 264)

Comment 16: In Figure 4B, it would be helpful to add HPV33, HPV52 and HPV 58 above the graphs, or at least indicate which graph corresponds to which virus in the legend.

Response: The figure has been amended to include these changes.

=====

Reviewer #2

General opinion and comments:

Reviewer: The article by Li et al. "Rational Design of a Triple-Type, Non-Env 1 eloped Virus Vaccine by Compromising the Type Specificity of Human Papillomavirus" describes how they performed loop swapping for all the five surface loops of L1 pentamers between a group of three closely related HPV types to enhance the level of cross-type protection to a significant level, even though the neutralization titer is generally lower than the WT HPV VLPs. The data are well presented, and will offer insight to the relative importance of different surface loops in determining the type-specific neutralising epitopes for HPV vaccine. This work will be of interest to those in viral immunology area, and is well suited for a more specialized journal.

Major comments (numbers in reviewer section indicate lines in the original manuscript):

Comment 1: Fig.1 shows the conformation differences of the five surface loops between a few pairs of HPV types. These are backbone conformations, and for the most part, if the loop length is the same (such as 33/58 pair), the loop conformations are identical, with the slight differences being within the variation of the same capsomere of the same structure. However, even in this situation, the surface contour will show difference because of the different side chains between the two types. For example, the -TKEG- of 58 versus -TSDS- of 33 on HI-loop, the surface features of KEG- should differ from those of SDS-. When the loop has different length, the loop conformation and surface contour is expected to be different, as shown by several L1 structured published from multiple groups. Such expected structural comparison results should be discussed. This consideration also can explain most of the efficacy effect of different loop swapping results.

Response: Thank you for this comment. We agree with the importance of structural comparisons for structure-based vaccine design. We performed structural comparisons in this study in three ways: (1) The root-mean-square deviation (RMSD), which measures the differences between two sets of atoms, was used to quantify the overall similarity by calculating the C α of loops between two specified types of L1 when their entire pentamer structures were superimposed. (2) Side-chains of comparable amino acid residues were compared in structure view in terms of their position and rotamers. (3) Surface contours of the compared regions were generated to detect any similarities in surface morphology. Please refer to the updated Figure 1C, where we now include information pertaining to the side-chain structure and sequence alignment of loops. As mentioned by the reviewer, the sequence length will significantly affect structural similarity. As suggested, we highlighted this consideration in the structural comparison results and analysis, "It should be noted that the sequence length can sometimes affect structural similarity, in case of which local loop structure should be examined along with RMSD calculation". (Pages 7-8, lines 121-123). Even though there are length differences between equivalent loops of the various types, in some cases, the RMSD can be used to reflect less conformation variation between the loops of genetically close types with sequence length differences than between loops of distant types with less of a discrepancy in sequence length. For example, the mean RMSDs

for the FG loop of HPV52 and -58 (with only two residues difference in length) is 0.52 Å, whereas the mean RMSDs for the BC loop of HPV58 and -59 (with only one residue difference in length) comes to 3.00 Å (Figure S4).

It will also be better if the pairwise loop sequence comparison are shown right below each Figure 1C panels. Another concern with such comparison is using the loop detached from the entire pentamer. Does any of these loops show larger or smaller difference when superimposed based on an entire pentamer between the two types?

Response: As suggested and as indicated in the aforementioned response, we have added sequence alignments to the updated Figure 1c. A structural comparison of loops from different L1 types (depicted in Figure 1c) was carried out with the entire pentamer structures superimposed. The relevant information has been added to the figure legend (Page 44, lines 846-848).

Comment 2: The aforementioned point is also reflected in Fig. 5, the determination of the chimeric pentamers: H58-33BC and H58-33BC-52HI to compare with the replaced loops, and concluded that BC loops of both H58-33BC and H58-33BC-52HI have the tips orientated ~7 Å and ~5 Å, respectively, away from the corresponding loop of the HPV58 pentamer. To show the different BC loop confirmations of the chimeric pentamers, the native pentamers of H58, 33, and 52 should be compared also in the same way.

Response: As suggested, we have added the distance measurements between the tips of the BC loops between HPV58 and -33, and between HPV58 and -52 in the revised manuscript, which reads as follows: “this value for the BC loops of HPV58/33 and HPV58/52 comes to ~4.5 Å and ~5.5 Å, respectively.” (Page 19, lines 309-310)

Comment 3: The H33 specific neutralizing mAb 4E5 was used to examine binding to the H58-33BC and H52-33BC-52HI chimera using EM. Because it binds to BC loop, it appears it was the right choice. It is unclear how the authors chose 4E5 for this study, do they already know it target H33-BC loop ahead of time? Did they test other mAbs that are specific for H33?

Response: We apologize that our decision to use 4E5 was unclear. The mAb 4E5 was chosen for cryoEM analysis because it showed the best binding activity against the chimeras among the HPV 33 specific neutralization mAb panel (Please see EC₅₀ data in the updated Table S4). To our knowledge, high binding affinity may facilitate structure determination of the VLP-Fab immune complex. As for the potential binding site, the redirected binding activity of the mAb 4E5 against H58-33BC indicated that its binding region may involve the H33-BC loop. We have not tested other mAbs that are specific for H33. A comprehensive characterization of these lead chimeras, including the finer epitope structural mapping, cross-neutralization antibody response and so on, will be carried out in future projects.

Comment 4: The loop swapping has been tried before to show the transfer of respective epitopes for a particular neutralizing antibody, although systematically extensively. This study tried swapping of all five loops between different types. Among the 5 surface loops, swapping EF loop has the least effect on transferring the cross-protection. Does that mean EF is the outermost positioned and most flexible loop to be a good neutralizing epitope? It would be good to add some discussion regarding this, and the differences of other loop transferring effect for different type.

Response: As suggested, we have included some comments regarding the loop transferring effect in the Discussion section, as follows: “Our data show that different loops have variable effects on transferring cross-protection between different groups (Figure 3); this could be primarily caused by their inconsistent antigenic

determinants for type specificity. Still, additional structural information of the chimeric VLPs needs to be considered to rationalize how to select particular loops for chimeras with high cross-protective efficacy. We also noticed that, among the five surface loops, there is no EF loop-exchange constructs for any of the lead triple-type candidates among the four groups (Group-1, -2, 3, -5 and -6) (Figure 3), even though an increased heterotypic neutralization could be observed in EF loop-swapped constructs; i.e., the anti-HPV58 antibody titer of H33-58EF was ~2- to 3-log higher than that found following immunization with WT HPV33 VLPs (Figure. 2c). One possible explanation is that the EF loop is not a major component in determining HPV type-specificity. Alternatively, it is possible that the EF loop displays higher flexibility than the other loops—as assumed by its relatively lower resolved density than the other loops in several L1 structures (Figures S2 and Supplementary Figure 11a)—and thus has a lower chance of providing a similar heterotypic functional area through EF loop exchange.” (Pages 21-22, lines 358-370)

Comment 5: It’s hard to guess the paper’s content from the current title. Title needs to be changed to reflect more on the actual work.

Response: As suggested, we have modified the title. The title now reads “Rational Design of a Triple-Type Human Papillomavirus Vaccine by Compromising Viral-Type Specificity”.

Minor comments (numbers in reviewer section indicate lines in the original manuscript):

Comment 6: Page 17, line 356 <5A-5C and S12> Where is S12?

Response: This has been corrected. (Page 19, line 323)

Comment 7: Page 16, line 332: <at levels comparable with that achieved by mixing three WT HPV VLPs at three times the dosage (versus one dose with the chimeric VLPs; Figure 3F)> Where is Figure 3F?

Response: Thank you for pointing out this error. This has been corrected to “Fig. 4b”. (Page 18, lines 301-302)

=====

Reviewer #3

General opinion and comments:

Reviewer: Chimeric VLP were designed in this work to be used to develop a better HPV vaccine that will prevent infection from a wider group of HPV serotypes and can be administered in a normally sized dose made with standard manufacturing practices. The proposed approach is based on a hypothesis that cross-type protection against HPV infection could be achieved by creating a chimeric HPV VLP. There is an impressive amount of work that has gone into the manuscript. The most compelling result is that all of the chimeric HPV33/58 VLPs showed improved heterotypic antibody responses, and this occurred without changing the elicitation of neutralization antibodies against their own backbone types. This finding was reinforced by the low ED50 values. However it was disappointing that cross-protective immunities cannot be elicited by proteins engineered using genetically distant types. More importantly though, unfortunately, there is a major problem with the basic premise and some smaller concerns.

Response: We thank the reviewer for recognizing the merit of our work. As reported, HPV neutralizing antibodies were shown to recognize epitope residues located on several surface loops of the L1 pentamer, which is a major reason why protection against HPV infection is highly type-restricted in terms of sequence diversity and structural variation among different types. Several studies using chimeric L1 proteins engineered among genetically distant HPV types elicit little type-cross neutralizing efficacy (*Christensen, Cladel et al.*

Virology, 2001), indicating that generating immunogens with high and broad anti-HPV potency will be a challenging task. Our previous work revealed that the binding of type-specific neutralizing antibodies can be redirected to genetically close HPV types by swapping only a few key epitope residues (Li, Wang et al, *mBio*, 2017), and this inspired our plan to develop a strategy to design cross-protective antigens of HPV focusing on phylogenetically close types. In this study, the structural differences and ancestral relationship among multiple HPV types was inspected, and used to guide this novel generation of candidate chimeras among close HPV types. In sum, our chimeras successfully elicit high cross-protective efficacy while maintaining protection against base type.

Major concerns (numbers in reviewer section indicate lines in the original manuscript):

Comment 1: Although it seems logical that there is a correlation between structural variation and evolutionary distance among HPV types, the results presented here do not support this basic premise. Using the calculated RMSD as a basis to support this hypothesis is flawed because the structural differences of L1 loops are insignificant. For example the mean RMSDs (for the variable region of paired genotypes) 0.66 Å (HPV33/58), 0.74 Å (HPV58/52), 0.94 Å (HPV11/16) and 2.02 Å (HPV58/59) do not support an increasing difference in structure. In fact quite the opposite is established from these RMSD values; this range actually supports a conclusion that the structural differences are negligible. To provide context, a hydrogen bond length is 2.2-3.2Å and the resolutions of the maps are 2.9 and 2.75Å.

Response: We thank the reviewer for giving us the opportunity to address this concern. The structures of the HPV major capsid protein, L1, have been well characterized in the past, with studies showing that the L1 monomer presents as a canonical, eight-stranded β -barrel, with the strands joined by six loops, five of which are located on the surface of the L1 pentamer, constituting a T=7 icosahedral capsid with 72 copies. The core β -strand structures of various L1s share essentially identical conformations; however, the surface-loop regions, which are believed to be associated with type specificity and viral antigenicity, differ significantly in conformation (Chen et al. *Molecular Cell*, 2000; Bishop et al. *J. Biol. Chem.* 2007; Lee, Brendle et al. *J. Virol.* 2014; Guan, Bywaters et al. *Virology*. 2015; Li et al. *Structure*, 2016; Li et al. *mBio*, 2017). In our present study, we correlated, for the first time, the structural differences among the L1 surface loops with evolutionary distance, as quantified by the mean RMSD of loop C α atoms and alignment of L1 sequences, and used a structure-based rationale to design cross-type HPV vaccines. As for how much the RMSD value means in terms of structural differences, we have included in the legend to Figure 1 information pertaining to the side-chain structures and sequence alignments of loops. We show that the structural differences in these loops increase with evolutionary distance (see from top-down in the updated Figure 1c). Thus, the RMSD range of 0.6 Å to 3 Å for the surface loops of the various types is reflected in the differences in structural diversity. Although the crystal structures used for comparison are at a resolution ranging from 2.0 Å to 3.5 Å, the electronic density maps of the surface loop regions were well-resolved and were used to build an atomic model, with prior knowledge of stereochemistry (Chen et al. *Molecular Cell*, 2000; Bishop et al. *J. Biol. Chem.* 2007; Li et al. *Structure*, 2016; Li et al. *mBio*, 2017). By virtue of these atomic models, we can confidently identify many hydrogen bonds (of length 2.2 Å to 3.2 Å) and make a structural comparisons at the atomic level. For example, although the mean RMSD (as low as 0.96 Å) for the BC loop of HPV58/33 is smaller than a hydrogen bond length, the tip of one loop orientates at least 4.5 Å away from the corresponding tip of the loop of another type, and this is a non-negligible difference in molecular structure.

Comment 2: This fallacy is reported again in Sup fig 6 where RMSD differences range from 0.1 - 0.6Å, a range that establishes no significant difference between structures.

Response: We believe that the Sup Fig 6 to which the reviewer is referring is actually Sup Fig. 4. While we agree with the reviewer that the RMSD range is not significant, this is not a “fallacy”. As we detail in response to the major concern in comment 1, these ranges are reported from well-resolved electronic density maps. Please also refer to the description in the main text, “In the comprehensive structural comparisons, although L1 proteins of different HPV types share almost identical structures in the core region (RMSDs of C α between any two in the range of 0.22 Å to 0.59 Å), the surface variable region, including the five loops, displays obviously different conformations among the L1 structures (RMSDs of C α from 0.53 Å to 2.06 Å; Supplementary Table 1).” (Page 8, lines 123-127)

Comment 3: Also Line 341-345. There are no remarkable HI loop differences in the structure of the main-chain carbon ...from the manuscript : “there were distinct differences in the surface contours of the (Figure 6C, lower right and Supplementary Figure 11B).” Figure 6C is the wrong figure to show this. SFig11B shows no significant differences in contour.

Response: We have incorrectly cited Figure 6c instead of Figure 5c. In addition, Sup Fig. 11b, which shows the models of the HI loops and their corresponding electronic density maps, should be cited earlier in this sentence. This sentence now reads, “For the HI loop, although we noted no remarkable differences in the structure of the main-chain carbon atoms in H58-33BC-52HI, HPV58, and HPV52 (Figure 5b and 5c, lower left, and Supplementary Figure 11b), there were distinct differences in the surface contours of the region bearing the HI loop for H58-33BC-52HI as compared with HPV58 and HPV33 but not for HPV52 (Figure 5C, lower right).” (Page 19, lines 310-314) Sup Fig. 11b (renumbered as Sup Fig. 11c in the updated Sup. Fig.11) shows the electronic density map for the HI loops, not the surface contours.

Comment 4: To be very clear, the structural differences do not support a conclusion that genetically closer types share more structural similarity than distant ones. Thus there is no “strong evidence” based on structural differences to explain the cross protection achieved by HPV vaccines against infection with a limited number of phylogenetically related HPV types. (line 158-160)

Response: In this study, our conclusions are based on several aspects. First, as indicated in our response to comment #1, the mean RMSDs were calculated to reveal the average distance between C α atoms of all residues on the loop(s) or the surface variable regions within compared types, which describes the overall structural variations in particular region(s). Then, the correlation between the mean RMSDs of the variable regions and phylogenetic distances between multiple pairwise types demonstrate that genetically closer types share more structural similarity than distant ones. Second, our data from the immunogenicity assays shows that close-type engineered chimeras, i.e., HPV33/58, more readily generate cross-protective sera than distant-type engineered chimeras, i.e. HPV58/59 (Figure 2 and Supplementary Figure 7). Third, employing the same vaccine strategy, we acquired a further four triple-type chimeric candidates (groups 2, -3, -5 and -6) that have high cross-protective neutralization titers against infection from three related HPV types (Supplementary Figure 9). Collectively, these three points suggest that the successful type-cross HPV vaccine derives from a better understanding of the structural and ancestral relationships among multiple HPV types. In this analysis, eight HPV L1 crystal structures were used: we acknowledge that more crystal structures would help to confirm a strong structure–evolution correlation. Thus, as suggested, we have toned down the wording, and rephrased the sentence as, “Therefore, our analysis, showing a correlation between structural variation and evolutionary distance among HPV types, provides primary structural evidence to explain the cross protection achieved by HPV vaccines against infection with a limited number of phylogenetically related HPV types”. (Page 9, lines 144-147)

Comment 5: Additionally, structural conservation with sequence divergence is a common tenet of structural virology. The concept that genetically closer types share more structural similarity than distant ones, particularly on the outer surfaces is not supported.

Response: We acknowledge that the common tenet of structural virology indicates structural conservation with sequence divergence. In the case of HPV structural virology, the L1 capsid has a conserved rigid structure for the core β -barrel framework; albeit, with variable sequences among different types. However, the outer surface region comprises relatively flexible loops, which are structurally variable and associated with type-specificity, antigenicity and protective immunogenicity (*Chen et al. Molecular Cell, 2000; Bishop et al. J. Biol. Chem. 2007; Li et al. Structure, 2016; Li et al. mBio, 2017*). In this study, the structural similarities of the loops of the different types of L1 crystal structures were inspected at the atomic level and quantified (mRMSD calculation) (the updated Figure 1C), revealing a correlation between structural differences and evolutionary distance (L1 sequence alignment). Cross-protection efficacy of the chimera was achieved with loop swapping on pairs that are close in type but this was not achieved for distant pairs, which supports our concept. These suppositions were further verified in four additional triple-type HPV chimeras designed in the same manner (Figure 3e).

Other concerns (numbers in reviewer section indicate lines in the original manuscript):

Comment 6: It is unclear what the antibody binding experiments add to our knowledge of HPV antigenic capsid structure. A single point mutation can abrogate binding of a neutralizing antibody, so it is no surprise that a chimeric loop also affects MAb binding.

Response: Unlike regular mutagenesis assays used for epitope mapping, antibody binding experiments were used to test whether the cross-type antigenicity was successfully redirected to the chimera by targeting the type-restricted epitope; this reflected the resultant cross-type immunogenicity after immunization *in vivo*.

Comment 7: H33-58HI and H58-33BC capsids are quite heterogeneous. Are they as stable as the others and does this have any bearing on their immunogenicity?

Response: The H33-58HI and H58-33BC chimeras are as stable as wild-type VLPs and other chimeras in terms of their purification, VLP assembly and storage processes, and display similar properties (Supplementary Figure 6). The high redirected cross-type antigenicity may be the reason for their relatively excellent cross-protective immunogenicity.

Comment 8: The authors state that residue substitutions on the HPV58 BC and HI loops change the orientation and conformation of the pentamer - this is overstated, somewhat incredible, and not shown in the figure cited. Any changes conferred by point mutations are typically small, likely limited by the side chain movement, but could affect local patch surface charge depending on the substitution. Therefore the conclusion is also overstated as no significant structural alteration has been demonstrated.

Response: We agree that this has been overstated. We have rephrased the sentence as: "These structural comparisons indicate that residue substitutions on the HPV58 BC and HI loops change the orientation and conformation of local regions to closely resemble that of HPV33 and HPV52, respectively". (Page 19, Lines 314-316)

Comment 9: For the cryo-EM maps, a central section of each map should be shown to verify the strength of the Fab density relative to the capsid density.

Response: The central section of each map is now included in the new Supplementary Figure 14.

Comment 10: At what sigma level are the surface contoured maps displayed? The capsomers are disconnected. Also, none of the maps appear to be 10-12Å resolution. According to the Methods, Relion was used to classify. These classes should be shown in Supplemental to address the issues of heterogeneity.

Response: All three cryoEM density maps are displayed at 2σ (sigma) for surface contouring. As suggested, the 2D classification results have been included in the new Supplementary Figure 14.

Comment 11: It is unclear whether gold standard procedure was followed throughout. The Auto3DEM version cited does not automatically split the data sets prior to reconstruction, which would mean that a FSC with cutoff of 0.5 should be used and not 0.3. Furthermore, auto3DEM can sometimes overestimate resolution, which could also be addressed by including central sections of the cryo-EM maps.

Response: Our apologies for the confusion. We used the latest version of Auto3DEM (4.05.2), as mentioned in Supplementary Table 8, which applied the gold standard. Because there were too few particles (~700-900 particles) involved in the final reconstruction and to avoid an over-estimation of resolution, which sometimes happens in the Auto3DEM program, we adjusted the FSC cutoff value from 0.143 to 0.3; this cutoff value of 0.3 has also been used in a previous study (*Guan et al. Virology, 2015*). As suggested, we included the central sections of the cryoEM maps in the new Supplementary Figure 14.

Comment 12: Resolutions for the complex maps are too poor to conclude that the Fabs have identical interactions.

Response: The description has been toned down, "... the 4E5 Fab binding orientation and footprints covering both the BC loop and the EF loop are similar for all three VLPs-Fab complexes" (Pages 20-21, lines 341-343).

Comment 13: Fig S13, Why does an FSC at 0.3 ensure resolution reliability of a reconstruction with only 100 particles? With the heterogeneity visible in the micrographs it is incredible that 10-12Å was attained with only 100 particles.

Response: Sorry for the typo. "... as only 100 particles were included in the final reconstructions" should be "... as only ~700-900 particles ..." (SI, Page15, line98). Please refer to "Number of particles used for reconstruction" in the updated Supplementary Table 8.

Comment 14: Icosahedral symmetry axes need to be added to the FAb-casomer panels (C) to provide context. Otherwise the angle seems arbitrary.

Response: The axes have been added, as suggested. Please refer to the updated Supplementary Figure 13.

Comment 15: In both Fig 6 and Fig S13 the roadmap needs to be described to explain to a reader how a 3-D map is rendered in 2-D, how were the contoured footprints were projected (and their significance), what section of the virus surface is displayed, and other information. Additionally, the Roadmap program should be properly cited in the legends.

Response: We apologize for the oversight in our citations. As suggested, detailed information has been included in the figure legends: "Fig. 6e Close-up view of the roadmaps of HPV33:4E5 (left), HPV58-33BC:4E5 (middle) and HPV58-33BC-52HI (right) surfaces, projected by an area surrounding a 5-fold vertex defined with a polar angle θ range of 46° to 58° , and a polar angle ϕ range of 68° to 86° . Color scheme for surface residues are according to surface loops (BC, amber; DE, orange; EF, yellow; FG, green and HI, Cyan). Blue

contour lines were drawn according to the projection of the Fab-4E5 entity with a density greater than 1 and within a radial section of the VLP-4E5 complex ranging from 310 Å to 360 Å. The icosahedral symmetry axes are labeled.” (Page 49, lines 922-928)

“Fig. S13d Global view of the roadmaps of HPV33:4E5 (left), HPV58-33BC:4E5 (middle) and HPV58-33BC-52HI (right) surface, projected by an area surrounding a 5-fold vertex defined with a polar angle θ range of 40° to 76°, and a polar angle ϕ range of 60° to 120°. Color scheme for surface residues are according to surface loops (BC, amber; DE, orange; EF, yellow; FG, green and HI, Cyan). Blue contour lines were drawn according to the projection of the Fab-4E5 entity with a density greater than 1 and within a radial section of the VLP-4E5 complex ranging from 310 Å to 360 Å. The icosahedral symmetry axes are labeled.” (SI, page 15, lines 104-111)

Comment 16: Roadmap use should be described (and cited) in the Methods. What conclusions are drawn from the projected contours of the Fab density? How was this projection done? Why are the surface residues not labeled?

Response: Roadmap is an excellent method to map the epitope residues, even with low resolution complexes. In this study, we mapped the rough footprints of 4E5 on the surface of HPV33 VLP and two chimera VLPs, which indicated similar binding sites for the chimeras involving the BC loop and the HI loop. As suggested, the detailed methods have been added to the Experimental Procedures: “The crystal structures of HPV33, HPV58-33BC, HPV58-33BC-52HI were fitted into the corresponding cryoEM density map using the tool “fit in map” in Chimera, and were then plotted onto stereographic spheres using RIVEM (Radial Interpretation of Viral Electron Density Maps) (Xiao et al. *Journal of structural biology*, 2007), where the polar angles θ and ϕ represent latitude and longitude, respectively. The Fab density was projected as contour lines onto the sphere, which is presented as a stereographic diagram.” (Pages 35-36, lines 593-597). Because low-resolution cryoEM structures were used for roadmap rendering, only rough regions (at loop level) should be depicted for Fab binding. Thus, surface residues were not labelled.

Comment 17: Discussion line 420 needs modification : “feasible approach, based on the knowledge of the intrinsic connection between structural diversity and evolution”

Response: As suggested, we detailed the feasible approach for a rationale for structure-based cross-type vaccine design: (1) Structure similarity (where the structure is available) and evolutionary distance analyses were performed to define type groups that could be incorporated into one chimera; (2) Chimeric loop-substituted mutants created between two most closely related types were initially generated to determine candidate(s) that would elicit an ideal double-type immunization response; constructs where swapping occurred between more structurally similar loops were prioritized, as only limited constructs could be generated and assayed for immunogenicity; (3) The remaining four loops of the double-type candidate(s) were then independently exchanged to the corresponding loop of a third genetically close type, and immunogenicity was evaluated to achieve the optimal triple-type chimeric VLPs; again, constructs where swapping had occurred on more structurally similar loops were prioritized. (Page 24, lines 393-403)

Comment 18: Influenza is not an appropriate ssDNA virus to reference in regard to lack of mutational drift.

Response: The sentence now reads “... Although HPV sequence diversity is not as high as that of RNA viruses, such as the human immunodeficiency and influenza viruses, or single-stranded DNA viruses, with extensive structural information, the method provided...” (Page 24, lines 403-405).

Reviewers' Comments:

Reviewer #1:

Remarks to the Author:

I'm satisfied with the changes in the manuscript that were made in response to my comments.

Reviewer #2:

Remarks to the Author:

The authors have made significant efforts to address the referee's concern to improve the manuscript, which has clarified a few of the concerns of mine and other referees and made the technical analysis part more solid. I think this manuscript has presented a great breadth of data covering immunology, structural biology, and virology. My lingering reservation since the first round review is that the core part of the work centered around the loop swapping on the capsomere (VLP) surface and the characterization of their immune response and neutralization, and the results did not show surprises from prior published work from multiple groups in that surface loops determine type specificity, and cross-protection (through neutralizing test) was limited to genetically closely related types, but not extended to more distantly related HPV types by such loop swapping.

Reviewer #3:

Remarks to the Author:

Although some concerns have been appropriately addressed, a major issue with the presentation of this work is a misconception of statistical significance on which main conclusions are based. Unfortunately the authors offered rebuttal instead of correcting the misinformation.

The limitations and degree of error involved in deriving root-mean-square-distance (rmsd) between pairs of equivalent atoms is well documented. The main issue here with rmsd is the statistical significance of rmsd differences. There is no statistical difference between RMSD values of 0.6, 0.7, and 0.9Å. The text and conclusions have not been adapted accordingly. The authors did not establish a conclusive link between evolution of the sequence and a predicted change to structure based on RMSD values, since the values reported have no significant difference. The rebuttal offered is not acceptable.

The problems of the misconception are further illustrated in Figure 5 C which shows quite well the differences in contour that can be seen when comparing maps that are closely related. If one observes the non-colored bumpy protrusions on the compared surfaces, it can be realized that there are just as many presumed differences between each map (in regions of identical sequence identity) as there are in what areas are highlighted in color to indicate a sequence difference. These maps are not significantly different in structure. Structural differences have been overstated and misrepresented in the presentation of the work.

Response to Reviewer Comments on the manuscript

Manuscript ID: NCOMMS-17-34032-T

We would like to thank all the reviewers for their constructive comments, which greatly improved our manuscript. We have addressed all the comments of the reviewers with appropriate additional reanalysis, summarized as follows:

- (1) The RMSD calculations throughout the manuscript have been removed. Figure 1 has been updated. The former Supplementary Figure 4, and Supplementary Tables 1,2 and 3 have been removed.
- (2) Figure 5 has been revised to pairwise surface comparisons instead of highlighted coloring on specific residues to avoid interpretation bias.
- (3) As requested, we have rephrased all of the statements considered to be overstated.

=====
Reviewer #1 (Remarks to the Author):

I'm satisfied with the changes in the manuscript that were made in response to my comments.

Response: We thank the Reviewer for their positive comments, which encouraged us to translate our cross-type vaccine candidate into clinical practice, with the aim to eradicate high-risk HPV infection in humans.

=====
Reviewer #2 (Remarks to the Author):

The authors have made significant efforts to address the referee's concern to improve the manuscript, which has clarified a few of the concerns of mine and other referees and made the technical analysis part more solid. I think this manuscript has presented a great breadth of data covering immunology, structural biology, and virology. My lingering reservation since the first round review is that the core part of the work centered around the loop swapping on the capsomere (VLP) surface and the characterization of their immune response and neutralization, and the results did not show surprises from prior published work from multiple groups in that surface loops determine type specificity, and cross-protection (through neutralizing test) was limited to genetically closely related types, but not extended to more distantly related HPV types by such loop swapping.

Response: We thank the Reviewer for recognizing the merit of our work and for her/his suggestions for the future direction of our vaccine development. In our response to the first round of comments, we acknowledged that type-specific neutralization of HPV is known to correlate to surface loops by structure determination of the capsid-nAb immune complex in both our previous study (Li et al. *mBio*, 2017) and in the studies of others (Guan et al. *Viruses*, 2017; Guan et al. *Virology*, 2015; Lee et al. *J Virol*, 2015). As pointed out by the Reviewer, the limitation in this strategy derives from the structural and molecular determinants of HPV type-specific neutralization, where the structures of HPV L1 within evolutionarily closely related types may share a similar structure in both surface loops and overall structural context. Thus, subtle differences in type-specific epitopes can be resurfaced by loop swapping of genetically close types but not distant ones. Regarding your lingering reservation, please allow us to reiterate the novelty of our findings over previous

studies:

(1) Our structural analysis of the surface loops of the various HPV L1 types indeed demonstrates that structural similarities of surface loops are highly conserved within groups of genetically closely related types but are substantially distinct among types belonging to different groups (See the updated Fig. 1); it should be noted that this structure-based conclusion has been revised according to Reviewer #3's comment.

(2) Extensive immunogenicity assays on the chimeric VLPs with surface loop swapping revealed that cross-type protection comparable with wild-type protection can only be achieved among genetically closely related types and fails in the case of evolutionarily distant types.

(3) A rational design strategy for a cross-type HPV vaccine was concept-proofed across three types of HPV (HPV58, 33 and 52), and can be generally applied in other triplet-type chimeras (HPV16/35/31, HPV56/66/53, HPV39/68/70, HPV18/45/59). Such a triple cross-type VLP antigen is unprecedented in the literature.

(4) Our structure determination of the potent chimeras, H58-33BC and H58-33BC-52HI, indicated that substitutions of several different residues could recapitulate the conformational contours of the antigenic surfaces of the parental-type proteins on the engineered region of the chimeric protein via subtle changes to the amino acid side-chains (see the updated Fig. 5). These map differences in the updated structural comparisons might be evidence to show how these two chimeras provide B-cell receptor recognition with broader breadth during MHC presentation and subsequently confer cross-genotype protection when used to immunize mice or primates. Please refer to Pages 18-19, Lines 298-313.

For evolutionarily distant HPV types, the surface loops and the structural contexts need to be examined to elucidate a mechanism to create cross-type immunity: this will be the focus of our future projects. Nevertheless, our study paves the way for the development of an improved HPV prophylactic vaccine against all carcinogenic HPV strains.

=====

Reviewer #3 (Remarks to the Author):

Although some concerns have been appropriately addressed, a major issue with the presentation of this work is a misconception of statistical significance on which main conclusions are based. Unfortunately the authors offered rebuttal instead of correcting the misinformation.

Response: We thank the Reviewer for their comments. We acknowledge our misunderstanding, and have substantially corrected the manuscript.

Comment 1: The limitations and degree of error involved in deriving root-mean-square-distance (rmsd) between pairs of equivalent atoms is well documented. The main issue here with rmsd is the statistical significance of rmsd differences. There is no statistical difference between RMSD values of 0.6, 0.7, and 0.9Å. The text and conclusions have not been adapted accordingly. The authors did not establish a conclusive link between evolution of the sequence and a predicted change to structure based on RMSD values, since the values reported have no significant difference. The rebuttal offered is not acceptable.

Response: We thank the Reviewer for this insight. As recommended, we have removed the RMSD calculations. We suggest that the HPV pentamer crystal structures—of which there are few in number—may not warrant measures of statistical differences for precise correlations among L1 sequences and surface loop structures. Instead, we illustrate differences in the loop structure among various HPV types of L1 pentamers using superficial structures, which demonstrate the high similar surface-loop structure among evolutionarily

close types (one phylogenetic group), and the substantial structural variation among those classified as distant types due to underlying sequence diversity. In the superimposed structures of various L1 types, most of the surface loops, including BC, DE, FG and HI, show quite a similar main-chain trace on the surface loop models (Fig. 1b) within a group in the phylogenetic tree (Fig. 1a); for example, Group 1, constituting HPV58, 33 and 52, exhibits prominent differences when compared with Group 6 (comprising HPV18, 45, and 59). These structural variations in the surface loops among distant types are associated with the aa sequence diversity in terms of residue type and sequence length; in contrast, where there is higher sequence identity and/or more similarity in sequence length between equivalent loops of the various L1 types, their main-chain structures may share more similarity (Fig. 1b). These structural comparisons, based on phylogenetic analysis, suggest that we may acquire a type-specific “compromise” via surface loop swapping within a phylogenetic group; however, we still need to screen for the loops that will work well in a vaccine design. As mentioned in our response to the first round of reviewers’ comments, the predominant neutralization epitopes that target surface loops are conformational and may comprise not only key *different* residues that are type-restricted but also a supportive structural context attributed to the overall structure, which is conserved in phylogenetically close but not distant types. The structure variation profile in the intra- and inter-group of HPV L1 structures is supported by the immunogenicity assay on loop-swapping chimeras: i.e., close-type engineered chimeras (HPV33/58) more readily generate cross-protective sera than distant-type engineered chimeras (HPV58/59) (Figure 2 and new Supplementary Figure 6). To reshape the basis for our cross-type vaccine design, the text and conclusion have been revised accordingly. Please refer to the updated Fig. 1; Page 2, Lines 24-26; Page 6, Lines 91-95; Pages 7-8, Lines 119-132 and Pages 8-9, Lines 136-146.

Comment 2: The problems of the misconception are further illustrated in Figure 5 C which shows quite well the differences in contour that can be seen when comparing maps that are closely related. If one observes the non-colored bumpy protrusions on the compared surfaces, it can be realized that there are just as many presumed differences between each map (in regions of identical sequence identity) as there are in what areas are highlighted in color to indicate a sequence difference. These maps are not significantly different in structure. Structural differences have been overstated and misrepresented in the presentation of the work.

Response: To rectify this, and to better demonstrate the structural differences, we now compare the structural difference by pairwise superimposition of the BC loops and HI loops of various pentamer crystal structures in surface-rendering mode (new Fig. 5). To avoid overstating the differences, we describe only the *most* different regions of the surface contours: these are ascribed to side-chain conformations and may be associated with the antibody binding activity and immunogenicity of the mutant VLPs. The analysis shows that the main chains of the loops in question display no significant differences (Fig. 5b - 5d); however, the side-chain rotamers of the mutated residues are more likely to recapitulate the conformation of the target type when the loops are swapped, as indicated by fewer different areas on the superimposed surface contours of the BC (Fig. 5c) and HI loops (Fig. 5d) between the chimeric and target types than between chimeric and wild-type ones. Thus, the map differences observed from the updated structural comparisons could be evidence to verify how these two chimeras—H58-33BC and H58-33BC-52HI—redirect and target type-specific neutralizing antibody binding (for instance, HPV33 specific mAb 4E5). These two chimeras may provide B-cell receptor recognition with broader breadth during MHC presentation and subsequently confer cross-genotype protection when used for immunization in mice or primates. The text and depiction on the crystal structures of mutant L1 pentamers have been revised to convey this. Please refer to the updated Fig. 5 and Pages 18-19, Lines 298-313.

Reviewers' Comments:

Reviewer #3:

Remarks to the Author:

Removal of the RMSD values, but continuing to insist on the same conclusions does not correct the miss-interpretation of the structural differences shown in this otherwise stellar work.

The correct interpretation of the finding is one of structural conservation despite sequence diversity, which is an established mechanism of virus evolution. This is a significant and important finding of the authors. The presentation would be stronger to put RMSD back into the paper in support of this conclusion that remarkably, the virus conserves basic loop structure even with divergent genotype and significantly altered phenotype.

Unfortunately the concluding statements continue to proceed in the wrong direction with different illustrations in figure one to offer support of divergent structures, which are not supported by small loop movements (as established by the authors). The more appropriate conclusion is that the authors have established a significantly interesting phenotype variation based on minor loop movement with side chain identity changes.

To place the changes in context, consider that a single point mutation can alter host recognition.

Specifically, for clarification, the structures are conserved. The work presented here shows structural conservation with sequence variation. Furthermore, this is an admirable representation of this virus structural and evolutionary phenomenon.

Unacceptable (wrong) conclusions:

Line 93 "the structural distinctions evident between two phylogenetic groups"

Line 125: "...yet with prominent differences noted between the two groups." An acceptable interpretation might read: "yet with minor differences noted among the antigenic loops structures between the two groups."

Line 138: "showing high structural similarities of the L1 surface loops within a phylogenetic group but structural distinction across different groups" There should instead be a conclusion similar to the following: "showing high structural similarities of the L1 surface loops within a phylogenetic group and remarkably extending across different groups, with only minor variation in the antigenic loops. This type of conserved structure despite genotypic evolution illustrates the variability that can be induced, while conserving function."

Response to Reviewer Comments on the manuscript

Manuscript ID: NCOMMS-17-34032B

We thank Reviewer #3 for recognizing the merit of our work and helping reshape the structural analysis on virus evolution. We have fully addressed all the comments as suggested and revised our manuscript accordingly. To facilitate the navigation of this document, we have copied the comments verbatim in blue and typed our responses in black.

REVIEWERS' COMMENTS:

Reviewer #3 (Remarks to the Author):

Comment 1: Removal of the RMSD values, but continuing to insist on the same conclusions does not correct the miss-interpretation of the structural differences shown in this otherwise stellar work.

Response: We have reshaped the conclusion to correct the mis-interpretation of the structural differences and reincluded the RMSD calculations to support this updated conclusion, as suggested in the next Comment. The conclusions were revised as follows:

“We initially found that the L1 proteins of various HPV types shared an overall conserved structure in their core regions and even in their basic surface loop configurations, with phenotype variations between different HPV types able to be induced by minor structural movements of the surface loops. Our design is based on a rational understanding of the high structural conservation of L1 surface loops within phylogenetically close HPV types despite sequence variation.” (page 6, line 91-96)

“Collectively, the structural information suggests high conservation of the overall structures of L1 within a phylogenetic group of HPV, which even extends across different groups, despite minor variations in the outer antigenic surface loops.” (page 9, line 142-144)

“We show that the overall loop structures are conserved for all five surface loops among various types of L1 structures. Nevertheless, the variable sequence identity of individual genotypes induces small loop movements between types, as shown in the superimposed structures.” (page 8, line 126-128)

Comment 2: The correct interpretation of the finding is one of structural conservation despite sequence diversity, which is an established mechanism of virus evolution. This is a significant and important finding of the authors. The presentation would be stronger to put RMSD back into the paper in support of this conclusion that remarkably, the virus conserves basic loop structure even with divergent genotype and significantly altered phenotype.

Response: We agree with the Reviewer on the mechanism of virus evolution, i.e., virus maintains its function(s) through structural conservation but evolving through sequence diversity to combat host immunity. As suggested, we re-included the RMSD calculations to support this conclusion. Please refer to the new Supplementary Table 2 and the interpretation on RMSD:

“Furthermore, pairwise structural comparisons were carried out between NCS copies of L1 monomers from five HPV types; the resulting pairwise root mean squared deviations (RMSDs) were plotted to generate the mean RMSDs for both the core regions and the surface loops of L1 for the various genotypes (Supplementary Tables 2). In the comprehensive structural comparisons, both the core region structures and the surface variable regions—constituted by five surface loops among the various types of L1 structures—shared conserved structural configurations, with RMSD values of C α ranging from 0.22 to 0.59 Å and from 0.53 to 2.06 Å, respectively (Supplementary Table 2). The minor loop movements, particularly between types from group 1 and group 6 (Fig. 1b), slightly increased the RMSD values with respect to that of the relatively rigid core region, which was believed to associate with the various type-specific phenotype variations underlying virus phylogenetic evolution. Collectively, the structural information suggests high conservation of the overall structures of L1 within a phylogenetic group of HPV, which even extends across different groups, despite minor variations in the outer antigenic surface loops.” (page 8-9, line 132-144)

Comment 3: Unfortunately the concluding statements continue to proceed in the wrong direction with different illustrations in figure one to offer support of divergent structures, which are not supported by small loop movements (as established by the authors). The more appropriate conclusion is that the authors have established a significantly interesting phenotype variation based on minor loop movement with side chain identity changes.

Response: As suggested, we have revised the interpretation for Figure 1 to reflect the updated conclusion.

“We show that the overall loop structures are conserved for all five surface loops among various types of L1 structures. Nevertheless, the variable sequence identity of individual genotypes induces small loop movements between types, as shown in the superimposed structures. We also observe that the surface loops of types within group 1 or group 6 have quite similar main-chain traces, with minor differences noted among the loops between the two groups. The extent of loop movement between L1s are correlated with aa sequence diversity in terms of residue type and sequence length (Fig. 1b).” (page 8, line 126-132)

Comment 4: To place the changes in context, consider that a single point mutation can alter host recognition.

Response: Thank you for this suggestion. We have included the results of all single point-swapping mutations in HPV58 or HPV33 L1 protein:

“Furthermore, we questioned whether the cross protection afforded by these two chimeras was achieved by multiple simultaneous mutations. We generated eight single mutation chimeras of HPV33 VLPs (mutated to the HPV58 HI loop: H33-S350K, H33-D351E, H33-S352G, H33-E357D) and HPV58 VLPs (mutated to the HPV33 BC loop: H58-S80N, H58-N82T, H58-N84A, H58-V87L). The neutralization titers of the anti-sera of these chimeras were measured and we found that most of the single-point mutants (except for H58-V87L) could elicit heterotypic neutralization antibodies (Supplementary Fig. 6) to some extent; this indicated that every residue associated with type-restricted neutralization contributes to the cross-neutralization when substituted to the other type. Intriguingly, the H33-E357D chimera was able to induce a cross-type neutralization titer as high as that of

H33-58HI (Supplementary Fig. 6), possibly demonstrating that residue D377 of HPV58 L1 takes a critical role during the immune recognition process of HPV58." (page 14, line 224-234)

Comment 5: Specifically, for clarification, the structures are conserved. The work presented here shows structural conservation with sequence variation. Furthermore, this is an admirable representation of this virus structural and evolutionary phenomenon.

Response: Thank you for the encouraging comment about our work.

Comment 6: Unacceptable (wrong) conclusions:

Line 93 "the structural distinctions evident between two phylogenetic groups"

Response: We have corrected this. The sentence now reads "We initially found that the L1 proteins of various HPV types shared an overall conserved structure in their core regions and even in their basic surface loop configurations, with phenotype variations between different HPV types able to be induced by minor structural movements on the surface loops. Our design is based on a rational understanding of the high structural conservation of L1 surface loops within phylogenetically close HPV types despite sequence variation ". (page 6, line 91-96)

Line 125: "...yet with prominent differences noted between the two groups." An acceptable interpretation might read: "yet with minor differences noted among the antigenic loops structures between the two groups."

Response: This suggestion has been included. (page 8, line 129-130)

Line 138: "showing high structural similarities of the L1 surface loops within a phylogenetic group but structural distinction across different groups" There should instead be a conclusion similar to the following: "showing high structural similarities of the L1 surface loops within a phylogenetic group and remarkably extending across different groups, with only minor variation in the antigenic loops. This type of conserved structure despite genotypic evolution illustrates the variability that can be induced, while conserving function."

Response: This suggestion has been included. The description now reads, "Therefore, our analysis on the structure of HPV33/58/52/18/59 types demonstrated that the L1 surface loops showed high structural similarities within a phylogenetic group, which remarkably extended across different groups, with only minor variation in the antigenic loops. This type of conserved structure despite genotypic evolution illustrates the variability that can be induced while conserving function, and then provides preliminary structural information to explain the cross-protection achieved by HPV vaccines against infection with a limited number of phylogenetically related HPV types." (page 9, line 148-154)